# Gut uropathogen abundance is a risk factor for development of bacteriuria and urinary tract infection

Matthew Magruder[1], Adam N. Sholi [1], Catherine Gong[1], Lisa Zhang[1], Emmanuel Edusei[1], Jennifer Huang [1], Shady Albakry[1], Michael J. Satlin[2], Lars F. Westblade[2,3], Carl Crawford[4], Darshana M. Dadhania[1,5], Michelle Lubetzky[1,5], Ying Taur[6], Eric Littman[6], Lilan Ling[6], Philip Burnham[7], Iwijn De Vlaminck [7], Eric Pamer[6], Manikkam Suthanthiran[1,5] & John Richard Lee[1,5]*

The origin of most bacterial infections in the urinary tract is often presumed to be the gut. Herein, we investigate the relationship between the gut microbiota and future development of bacteriuria and urinary tract infection (UTI). We perform gut microbial profiling using 16S rRNA gene deep sequencing on 510 fecal specimens from 168 kidney transplant recipients and metagenomic sequencing on a subset of fecal specimens and urine supernatant specimens. We report that a 1% relative gut abundance of *Escherichia* is an independent risk factor for *Escherichia* bacteriuria and UTI and a 1% relative gut abundance of *Enterococcus* is an independent risk factor for *Enterococcus* bacteriuria. Strain analysis establishes a close strain level alignment between species found in the gut and in the urine in the same subjects. Our results support a gut microbiota–UTI axis, suggesting that modulating the gut microbiota may be a potential novel strategy to prevent UTIs.

[1] Division of Nephrology and Hypertension, Department of Medicine, Weill Cornell Medicine, New York, NY, USA. [2] Division of Infectious Diseases, Department of Medicine, Weill Cornell Medicine, New York, NY, USA. [3] Department of Pathology and Laboratory Medicine, Weill Cornell Medicine, New York, NY, USA. [4] Division of Gastroenterology, Department of Medicine, Weill Cornell Medicine, New York, NY, USA. [5] Department of Transplantation Medicine, New York Presbyterian Hospital – Weill Cornell Medical Center, New York, NY, USA. [6] Infectious Disease Services, Department of Medicine, Memorial Sloan Kettering Cancer Center, New York, NY, USA. [7] Department of Biomedical Engineering, Cornell University, Ithaca, NY, USA. *email: jrl2002@med.cornell.edu

Urinary tract infection (UTI) is one of the most common bacterial infections around the world, affecting over 150 million people yearly[1]. In the United States, it leads to over 10 million ambulatory visits and the economic burden is estimated to be greater than 2 billion dollars per year[2]. Kidney transplant recipients are a unique population to study the pathogenesis of UTI as they are frequently diagnosed with bacteriuria (positive urine culture without clinical symptoms) and UTI (positive urine culture with clinical symptoms of dysuria, frequency, urgency, or fever)[3]. Bacteriuria occurs in 40% of kidney transplant recipients within the first year after transplantation[4] and UTI occurs in 50% within the first three years after transplantation[5]. Asymptomatic bacteriuria alone can have significant consequences in this patient population, increasing the risk of pyelonephritis seven-fold in a study of 189 kidney transplant recipients[6]. In most cases, bacteriuria and UTIs are treated effectively with antibiotics, but in rare cases, they can lead to morbidity and mortality. Approximately 1–3% of UTIs are associated with bacteremia development[7,8], and UTIs occurring greater than 6 months after transplantation have been associated with increased risk of death and graft loss in a retrospective study of over 20,000 kidney transplant recipients[5].

The pathogenesis of UTI is complex. Contamination of the periurethral space by a gut uropathogen is considered the initial step, followed by colonization of the urethra and bladder[9]. Elegant studies illuminating the colonization steps have elucidated the role of pili and flagella in the successful propagation of uropathogens during the development of UTI[9]. Few studies, however, have investigated the initial phase of UTI development and the role that gut microbiota may have. In a pilot study of 26 kidney transplant recipients in which we performed serial gut microbiota profiling, we reported an association between gut Enterococcus abundance and Enterococcus UTIs[10]. In another recent case-control study of children with UTIs and without UTIs, the relative gut abundance of Escherichia coli was significantly higher in children with UTIs than children without UTIs[11]. Both of these studies assessed whether the gut microbiota is associated with UTIs at the time of UTI, but not whether the gut microbiota is a risk factor for development of bacteriuria and UTI.

The primary focus of this study is to investigate the association between the gut microbiota and the risk of developing bacteriuria or UTI. In a cohort of kidney transplant recipients at our transplant center, conventional urine cultures are performed at every clinic visit, allowing for a detailed record of bacteriuria in this cohort. We collect serial fecal specimens from 168 kidney transplant recipients within the first 3 months of transplantation and perform 16S rRNA gene deep sequencing on these specimens to evaluate the gut microbiota's association with future development of bacteriuria and UTI. In a subset of subjects with bacteriuria, we evaluate urine supernatant specimens via cell-free DNA sequencing and paired fecal specimens via shotgun metagenomic sequencing to assess the strain similarity, uropathogenic genes, and antibiotic resistance genes. We report that gut abundance of uropathogens is associated with future development of bacteriuria and UTI and that strain analysis establishes a close strain level alignment between species found in the gut and in the urine in the same subjects with bacteriuria.

## Results

**Characteristics of the kidney transplant cohort.** One hundred sixty-eight kidney transplant recipients provided a total of 510 fecal specimens in the first 3 months after transplantation. Fecal specimens were collected at post-transplant week 1, week 2, week 4, and week 12 and during episodes of diarrhea as well as during episodes of UTIs. Conventional urine cultures were performed at every routine clinic visit (twice weekly in the first month, weekly in the second month, every 2 weeks in the third month, and monthly up to 6 months), allowing for a comprehensive evaluation of bacteriuria in these subjects. 16S rRNA gene deep sequencing of the V4–V5 hypervariable region was performed on the 510 fecal specimens and the mean average ± standard deviation (SD) high quality classified reads were 18,179 ± 11,033.

Among the 168 kidney transplant recipients, 102 subjects developed bacteriuria (≥10,000 colony-forming units [cfu]/mL) (Bacteriuria Group) and 66 subjects did not in the first 6 months after transplantation (No Bacteriuria Group) (Supplementary Fig. 1). Differences in clinical and transplant characteristics are found in Supplementary Table 1. Female gender and older age were significantly associated with the Bacteriuria Group ($P < 0.05$). The five most abundant bacteria identified included: 36 subjects with Escherichia species (spp.) bacteriuria, 36 subjects with Enterococcus spp. bacteriuria, 20 subjects with Klebsiella spp., 44 subjects with Staphylococcus spp., and 19 subjects with Streptococcus spp. The full list of bacteriuria episodes and the organisms isolated are found in Supplementary Table 2.

**Gut abundance and respective bacteriuria group.** We evaluated the relative gut abundance of the five most abundant genera associated with bacteriuria (Escherichia, Enterococcus, Klebsiella, Staphylococcus, and Streptococcus) and its relationship to respective bacteriuria group within the first 3 months of transplantation (Fig. 1). The Escherichia relative gut abundance was significantly higher in the Escherichia Bacteriuria Group than in the No Escherichia Bacteriuria Group (Fig. 1a) ($P < 0.001$, Wilcoxon rank sum test), and the Enterococcus relative gut abundance was significantly higher in the Enterococcus Bacteriuria Group than in the No Enterococcus Bacteriuria Group ($P < 0.001$, Wilcoxon rank sum test) (Fig. 1b). The relative gut abundances of Klebsiella, Staphylococcus, and Streptococcus were not significantly higher in their respective Bacteriuria Group than in their respective No Bacteriuria Group ($P > 0.05$, Wilcoxon rank sum test) (Fig. 1c–e).

We further evaluated the relative gut abundance of Escherichia and Enterococcus in the cohort, stratified by gender and by older age (age ≥ 65). The relative gut abundance of Escherichia was not significantly different between male samples and female samples ($P = 0.16$, Wilcoxon rank sum test, Supplementary Fig. 2a) or between older age and younger age ($P = 0.86$, Wilcoxon rank sum test, Supplementary Fig. 2b). The relative gut abundance of Enterococcus was significantly higher in the female samples than in the male samples ($P = 0.007$, Wilcoxon rank sum test, Supplementary Fig. 2c) and the relative gut abundance of Enterococcus was significantly higher in the older age samples than in the younger age samples ($P = 0.02$, Wilcoxon rank sum test, Supplementary Fig. 2d).

**Gut abundance and the risk for bacteriuria and UTI.** In order to account for the timing of the relative gut abundance with relationship to bacteriuria development, we utilized a time-dependent Cox regression analysis to evaluate whether relative gut abundance of bacteria is a risk factor for future development of Escherichia and Enterococcus bacteriuria as these two genera were significantly different in the gut abundance analyses. To account for other clinical variables associated with bacteriuria, we analyzed clinical characteristics that we have previously analyzed[7] as well as the cause of end stage renal disease and calculated panel reactive antibody status in multivariable Cox regression analysis. We utilized a cutoff of 1% relative abundance as this was the

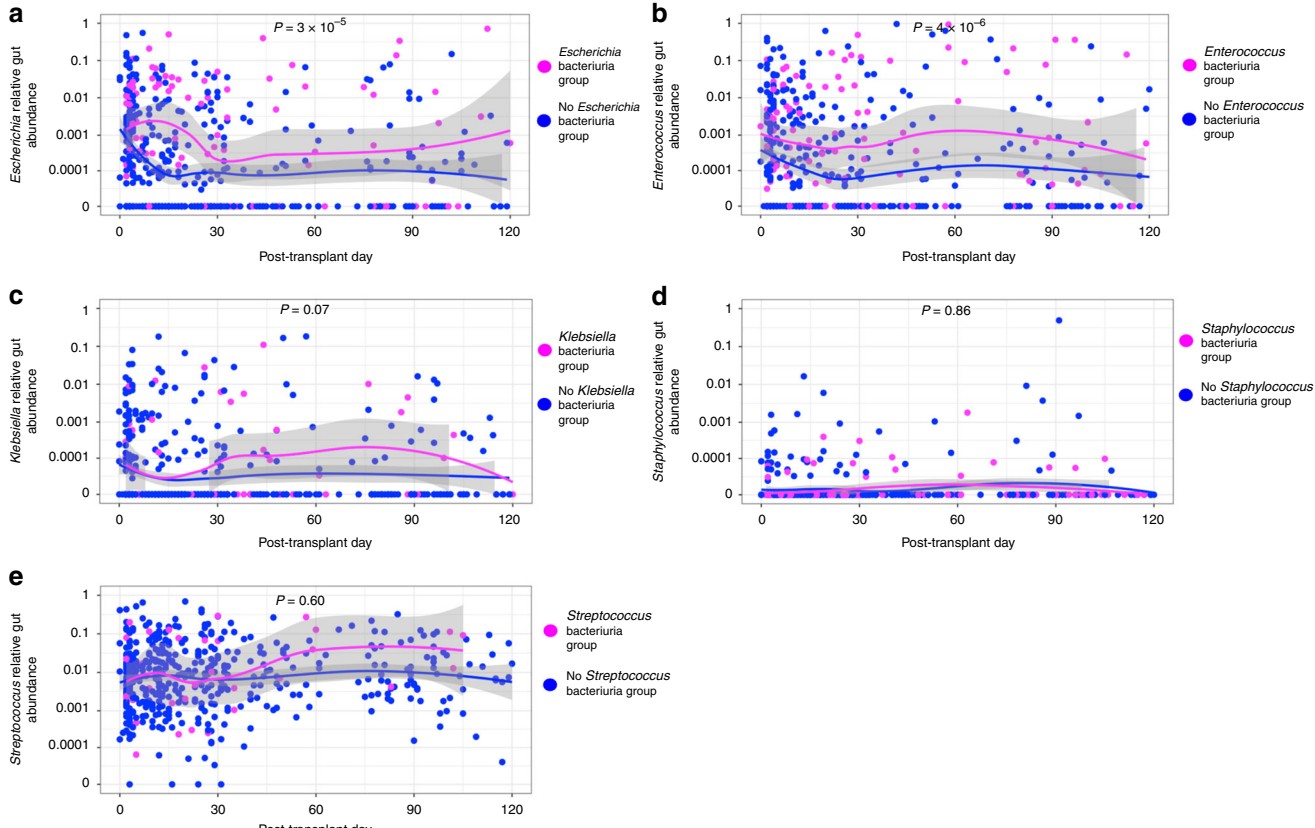

**Fig. 1** Temporal dynamics of relative gut abundances and respective bacteriuria. The relative gut abundance of bacteria is on the y-axis (logarithmic scale) and the post-transplant days stool specimens were collected are on the x-axis. The 510 specimens are each represented by a magenta-colored point reflecting a specimen belonging to the respective Bacteriuria Group and a blue-colored point reflecting the respective No Bacteriuria Group. LOESS curves with 95% confidence intervals (in gray) were created by group status. Comparison of relative gut abundances by group status was performed using a Wilcoxon rank sum test. **a** Temporal dynamics of *Escherichia* relative gut abundance following kidney transplantation by *Escherichia* Bacteriuria Group status. **b** Temporal dynamics of *Enterococcus* relative gut abundance following kidney transplantation by *Enterococcus* Bacteriuria Group status. **c** Temporal dynamics of *Klebsiella* relative gut abundance following kidney transplantation by *Klebsiella* Bacteriuria Group status. **d** Temporal dynamics of *Staphylococcus* relative gut abundance following kidney transplantation by *Staphylococcus* Bacteriuria Group status. **e** Temporal dynamics of *Streptococcus* relative gut abundance following kidney transplantation by *Streptococcus* Bacteriuria Group status. Source data are provided as a source data file.

upper limit in the locally weighted scatterplot smoothing (LOESS) curves of *Enterococcus* and *Escherichia* graphs in Fig. 1.

Using a univariate Cox Regression analysis with the relative gut abundance as a time-dependent covariate, we found that a 1% relative gut abundance of *Escherichia* was associated with future development of *Escherichia* bacteriuria (Hazard Ratio [HR]: 2.8, $P = 0.002$) (Table 1). In multivariate analysis that included the significantly associated univariate characteristics of female gender and cefazolin preoperative antibiotic prophylaxis, a 1% relative gut abundance of *Escherichia* continued to be associated with future development of *Escherichia* bacteriuria (HR: 2.8, $P = 0.002$) (Table 1). We also found that a 1% relative gut abundance of *Enterococcus* was associated with future development of *Enterococcus* bacteriuria (HR: 2.4, $P = 0.01$) (Table 2). In multivariate analysis that included the significantly associated univariate characteristics of deceased donor transplantation and delayed graft function, a 1% relative gut abundance of *Enterococcus* continued to be associated with future development of *Enterococcus* bacteriuria (HR: 2.3, $P = 0.02$) (Table 2).

We also analyzed the relationship between 1% relative gut abundance of *Escherichia* and future development of *Escherichia* UTI. In univariate analysis, a 1% relative gut abundance of *Escherichia* was associated with future development of *Escherichia* UTI (HR: 2.9, $P = 0.01$) and in multivariable analysis controlling for gender, a 1% relative gut abundance of *Escherichia* continue to

be associated with future development of *Escherichia* UTI (HR: 2.8, $P = 0.02$) (Supplementary Table 3). In univariate analysis, we did not find a significant relationship between a 1% relative gut abundance of *Enterococcus* and future development of *Enterococcus* UTI (HR: 1.8, $P = 0.28$) (Supplementary Table 4).

**Strain similarity between gut and urine strains.** To assess the relatedness of strains in the gut and in the urine, we evaluated 17 fecal specimens that were paired to 17 urine specimens in which urine cultures were positive for *Escherichia coli* ($n = 14$), *Enterococcus faecalis* ($n = 2$), and *Enterococcus faecium* ($n = 2$) (one urine culture was positive for both *E. coli* and *E. faecalis*) and in which relative gut abundance of *Escherichia* or *Enterococcus* was above two percent by 16S rRNA gene sequencing. The timing of the collection of stool specimens in relationship to the collection of urine specimens is shown in Supplementary Fig. 3. The 2% cutoff was chosen so that we would have at least an estimated 5X genome coverage of *E. coli*, *E. faecalis*, and *E. faecium* for strain analysis based on an estimated 40 million reads per sample and 50% alignment to human sequences.

The supernatants from these 17 urine specimens underwent single-stranded library preparation for cell-free DNA profiling[12]. Shotgun metagenomics sequencing was previously performed with an average mean ± SD depth of 55 ± 8 million reads[13] with

**Table 1 Gut microbiota abundance and future development of *Escherichia* bacteriuria.**

|  | Univariate analysis | | Multivariate analysis | |
|---|---|---|---|---|
| Characteristic | HR (95% CI) | *P* value | HR (95% CI) | *P* value |
| Age, Years | 1.0 (1.0–1.0) | 0.80 | | |
| **Female gender** | **7.4 (3.1–17.9)** | **7.4E−06** | **7.9 (3.3–18.9)** | **4.2E−06** |
| African American Race | 1.1 (0.5–2.3) | 0.78 | | |
| Diabetes mellitus | 1.2 (0.6–2.4) | 0.58 | | |
| Prior kidney transplant | 0.9 (0.3–2.2) | 0.75 | | |
| Cause of ESRD - DM | 1.1 (0.6–2.3) | 0.71 | | |
| Cause of ESRD - HTN | 0.6 (0.2–1.8) | 0.40 | | |
| PRA ≥ 80% | 1.3 (0.5–3.7) | 0.60 | | |
| Deceased donor transplantation | 1.4 (0.7–2.8) | 0.32 | | |
| Delayed graft function | 1.0 (0.4–2.5) | 0.95 | | |
| **Cefazolin preoperative abx** | **0.5 (0.2–1.0)** | **0.06** | **0.5 (0.2–1.0)** | **0.06** |
| TMP-SMX PCP prophylaxis | 2.1 (0.3–15.7) | 0.45 | | |
| Anti-thymocyte globulin induction | 1.7 (0.7–4.2) | 0.22 | | |
| Prednisone maintenance | 1.1 (0.6–2.3) | 0.71 | | |
| **1% *Escherichia* relative abundance** | **2.8 (1.4–5.4)** | **0.002** | **2.8 (1.5–5.5)** | **0.002** |

A Cox Proportion Hazard Model was used to assess the relationship between gut microbial abundance and future development of *Escherichia* bacteriuria. A 1% relative gut abundance of *Escherichia* was assessed as a time-dependent covariate. The hazard ratio (HR) with 95% confidence intervals (CI) is reported with the associated *P* value. Univariate analysis was performed with all of the characteristics. Characteristics that were significantly associated with *Escherichia* bacteriuria (*P* < 0.10) were further analyzed in the multivariate analysis. Bold text are the characteristics associated with future development of *Escherichia* bacteriuria. Cause of ESRD is ordinal data and the reference for the HR is Cause of ESRD —other. Source data are provided as a source data file
*DM* diabetes mellitus, *HR* hazard ratio, *HTN* hypertension, *PRA* panel reactive antibodies, *TMP-SMX* trimethoprim-sulfamethoxazole, *PCP Pneumocystis jiroveci*

**Table 2 Gut microbiota abundance and future development of *Enterococcus* bacteriuria.**

|  | Univariate analysis | | Multivariate analysis | |
|---|---|---|---|---|
| Characteristic | HR (95% CI) | P value | HR (95% CI) | P value |
| Age, Years | 1.0 (1.0–1.0) | 0.31 | | |
| Female gender | 1.7 (0.9–3.2) | 0.13 | | |
| African American Race | 1.2 (0.6–2.5) | 0.65 | | |
| Diabetes mellitus | 1.4 (0.7–2.8) | 0.29 | | |
| Prior kidney transplant | 1.0 (0.4–2.5) | 0.97 | | |
| Cause of ESRD—DM | 1.7 (0.8–3.5) | 0.16 | | |
| Cause of ESRD—HTN | 1.7 (0.7–4.1) | 0.25 | | |
| PRA ≥ 80% | 1.6 (0.6–4.6) | 0.36 | | |
| **Deceased donor transplantation** | **2.2 (1.1–4.2)** | **0.02** | **1.8 (0.8–4.1)** | **0.17** |
| **Delayed graft function** | **2.2 (1.1–4.6)** | **0.03** | **1.4 (0.6–3.5)** | **0.46** |
| Cefazolin Preoperative Abx | 1.3 (0.5–3.4) | 0.54 | | |
| TMP-SMX PCP Prophylaxis | 1.0 (0.2–4.2) | 0.99 | | |
| Anti-thymocyte globulin induction | 0.8 (0.4–1.6) | 0.51 | | |
| Prednisone maintenance | 1.1 (0.5–2.3) | 0.74 | | |
| **1% *Enterococcus* relative abundance** | **2.4 (1.2–4.8)** | **0.01** | **2.3 (1.2–4.6)** | **0.02** |

A Cox Proportion Hazard Model was used to assess the relationship between gut microbial abundance and future development of *Enterococcus* bacteriuria. A 1% relative gut abundance of *Enterococcus* was assessed as a time-dependent covariate. The hazard ratio (HR) with 95% confidence intervals (CI) is reported with the associated *P* value. Univariate analysis was performed with all of the characteristics. Characteristics that were significantly associated with *Enterococcus* bacteriuria (*P* < 0.10) were further analyzed in the multivariate analysis. In bold text are the characteristics associated with future development of *Enterococcus* bacteriuria. Cause of ESRD is ordinal data and the reference for the HR is Cause of ESRD—Other. Source data are provided as a source data file
*DM* diabetes mellitus, *HR* hazard ratio, *HTN* hypertension, *PRA* panel reactive antibodies, *TMP-SMX* trimethoprim-sulfamethoxazole, *PCP Pneumocystis jiroveci*

75% ± 12% (mean ± SD) of sequences aligned to human chromosomes and removed. The 17 paired fecal specimens underwent shotgun metagenomic sequencing with an average mean ± SD depth of 48 ± 6 million reads with 0.4% ± 0.9% (mean ± SD) sequences aligned to the human chromosomes and removed. Further details about the amount of urine DNA recovered and the estimated coverage of the *E. coli, E. faecalis*, and *E. faecium* genomes are found in the Supplementary Note 1.

We utilized StrainPhlAn[14] to assess the relatedness of the strains in the urine and in the stool. In brief, StrainPhlAn maps sequences from a specimen against the MetaPhlAn2 marker database[15] to produce a consensus-marker sequence for species in the sample. StrainPhlAn then utilizes MUSCLE[16] to align consensus sequences from samples to produce a multiple sequence alignment and utilizes RAxMl[17] to build the phylogenetic tree. Importantly, post-processing operations are utilized at each step to ensure high quality consensus sequences. In order to calculate differences at the base pair level, we utilized distmat from the European Molecular Biology Open Software Suite[18] which measures the number of single nucleotide variants (SNVs) per 100 base pairs.

Utilizing StrainPhlAn, we obtained 20 consensus strains of *E. coli* among all of the 34 urine and fecal specimens among which there were 6 pairs of *E. coli* urine and stool from the same subject. Figure 2a shows the related phylogeny of the *E. coli* consensus strains; 5 of the 6 *E. coli* urine strains were closely related to the *E. coli* strains from the fecal specimens from the same subject. Using the SNVs per 100 base pair measurement, the closest strains for Subject 54 Urine 2, Subject 36 Urine, Subject 41 Urine, Subject 106 Urine, and Subject 165 Urine were the strains from the subjects'

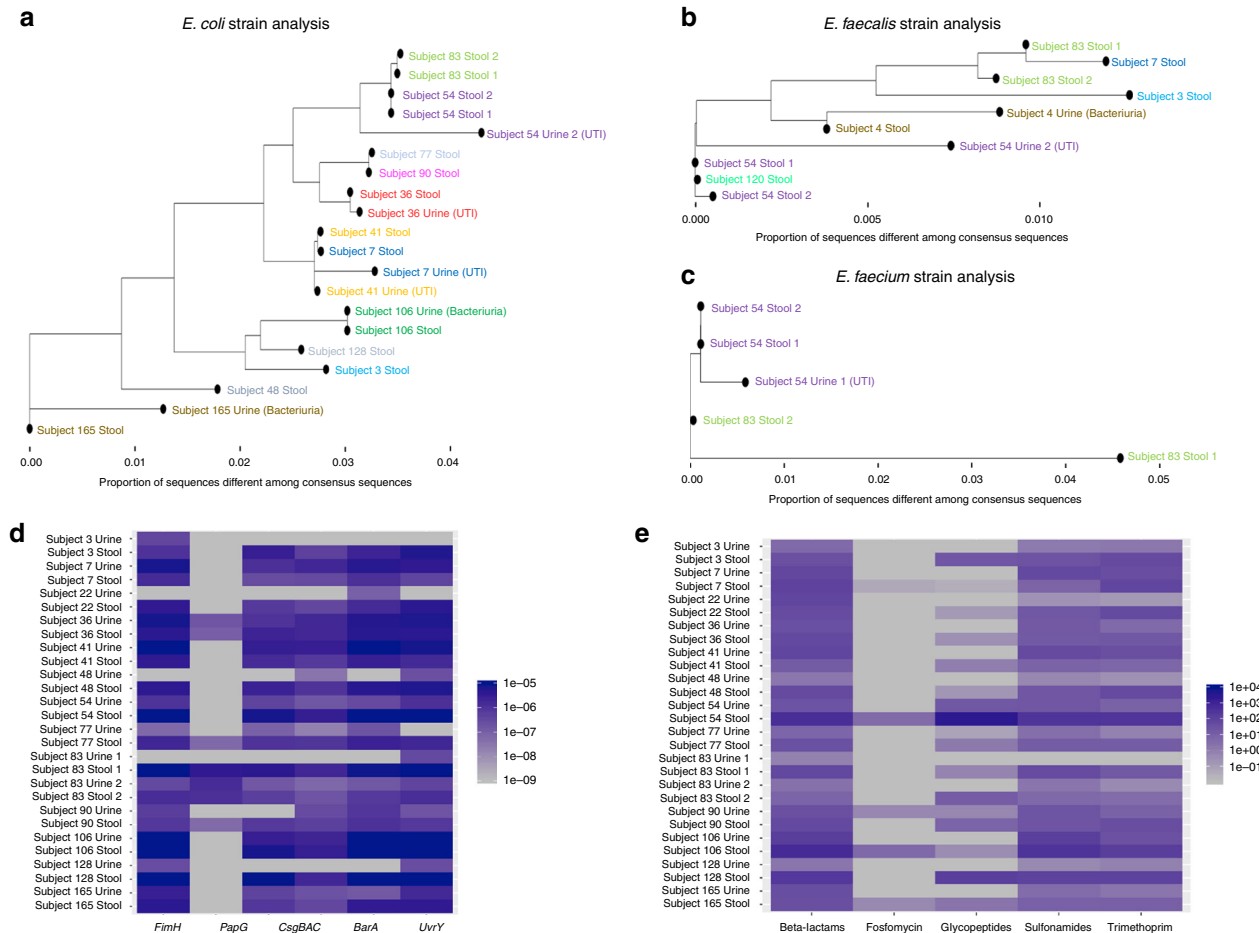

**Fig. 2** Strain analysis, uropathogenic genes, and antimicrobial resistance genes in paired urine-fecal specimens. **a** Among 34 urine and fecal specimens profiled, 20 consensus strains for *E. coli* could be constructed using StrainPhlAn. A phylogenetic tree was constructed based on the *E. coli* strain alignment (24 markers) and the proportion of sequences that are different between strains is noted on the *x*-axis. Each point represents an *E. coli* strain from a urine or fecal specimen with different colors representing different subjects. **b** Among 34 urine and fecal specimens profiled, 10 consensus strains for *E. faecalis* could be constructed using StrainPhlAn. A phylogenetic tree was constructed based on the *E. faecalis* alignment (200 markers) and the proportion of sequences that are different between strains is noted on the *x*-axis. Each point represents an *E. faecalis* strain from a urine or fecal specimen with different colors representing different subjects. **c** Among 34 urine and fecal specimens profiled, 5 consensus strains for *E. faecium* could be constructed using StrainPhlAn. A phylogenetic tree was constructed based on the *E. faecium* alignment (200 markers) and the proportion of sequences that are different between strains is noted on the *x*-axis. Each point represents an *E. faecium* strain from a urine or fecal specimen with different colors representing different subjects. **d** Bacterial genes were determined using HUMAnN2[19] and relative abundance was estimated for each of the following uropathogenic *E. coli* associated genes: *FimH*, *PapG*, *CsgBAC*, *BarA*, and *UvrY*. A heatmap was constructed with the uropathogenic genes on the *X*-axis and the *E. coli* associated urine specimens and paired stool specimens on the *Y*-axis. The abundance is colored by blue intensity, log scaled. **e** Antibiotic resistance genes were determined using Bowtie2[21] on the MEGARES antibiotic resistance database[22] and RPKM was estimated for genes that confer resistance to beta-lactams, fosfomycin, glycopeptides, sulfonamides, and trimethoprim. A heatmap was constructed with antibiotic resistance gene classes on the *X*-axis and the *E. coli* associated urine specimens and paired stool specimens on the *Y*-axis. The abundance is colored by blue intensity, log scaled. Source data are provided as a source data file.

own stool samples. The closest strain for Subject 7 Urine was the strain from Subject 41 Urine but very closely similar to the strain from Subject 7 Stool (Supplementary Table 5).

We obtained 10 consensus strains of *E. faecalis* among all of the 34 urine and fecal specimens among which there were 2 pairs of *E. faecalis* urine and stool from the same subject. Figure 2b shows the related phylogeny of the *E. faecalis* consensus strains and that both *E. faecalis* urine strains were closely related to the *E. faecalis* strains from the fecal specimens from the same subject. Using the SNVs per 100 base pair measurement, the closest strains for Subject 4 Urine and Subject 54 Urine 2 were the strains from the subjects' own stool samples (Supplementary Table 6).

We obtained five consensus strains of *E. faecium* among all of the 34 urine and fecal specimens among which there was 1 pair of

*E. faecium* urine and stool from the same subject. Figure 2c shows the related phylogeny of the *E. faecium* consensus strains and that the *E. faecium* urine strain was closely related to the *E. faecium* strain from the fecal specimen from the same subject. Using the SNVs per 100 base pair measurement, the closest strains for Subject 54 Urine 1 was the strain from the subject's own stool sample (Supplementary Table 7).

**Uropathogenic genes and antibiotic resistance genes**. We further analyzed the 14 paired urine and fecal specimens associated with *E. coli* bacteriuria for bacterial genes. Bacterial gene alignments were determined using HUMAnN2[19] and normalized to relative abundance using total sum scaling. We evaluated the following uropathogenic genes associated with UTI:[20] *FimH*, *PapG*, *CsgBAC*,

*BarA*, and *UvrY*. *FimH* is a gene that is part of the *fim* operon that encodes type 1 pili, allowing for attachment to the bladder epithelium; *PapG* is a gene that is part of the *pap* operon that encodes the P-type pilus, allowing for attachment to the bladder epithelium; *CsgBAC* is part of the operons that coordinates curli formation, which can facilitate biofilm formation; *BarA* and *UvrY* are part of the two component signaling systems, which regulate metabolic pathways and allow for adaptation in the urinary tract[20]. We found the presence of these genes in both paired urine and fecal specimens (Fig. 2d). Specifically, *FimH*, *CsgBAC*, *BarA*, and *UvrY* were found in both the urine and fecal specimens from the same subject in a majority of the 14 paired urine and fecal specimens, supporting the similarity of urine and gut *E. coli* strains from the same subjects. Five of the 14 urine samples were associated with symptoms and there were no significant differences in the abundance of any of these genes between the symptomatic *E. coli* UTI and asymptomatic *E. coli* bacteriuria ($P > 0.05$, Wilcoxon rank sum test), but there was a trend towards a higher abundance of *CsgBAC* and *BarA* in the symptomatic *E. coli* UTI than the asymptomatic *E. coli* bacteriuria ($P = 0.08$, $P = 0.08$, Wilcoxon rank sum test, respectively) (Supplementary Fig. 4a–e).

We also analyzed the paired urine and fecal specimens for antibiotic resistance genes. Antibiotic resistance genes were determined using Bowtie2[21] on the MEGARES antibiotic resistance genes database[22] and were normalized to reads per kilobyte per million (RPKM). We evaluated genes conferring resistance to the following classes of antibiotics: beta-lactams, fosfomycin, glycopeptides, sulfonamides, and trimethoprim. Evaluation using the MEGARES antibiotic resistant genes database[22] included 115 gene/operons in the beta-lactams class; 7 gene/operons in the fosfomycin class; 72 gene/operons in the glycopeptide class; 4 gene/operons in the sulfonamides class; and 11 genes/operons in the trimethoprim class (The full list of genes/operons is provided in Supplementary Data 1.) Beta-lactams, sulfonamides, and trimethoprim resistance genes were present in both urine and fecal specimens from the same subject in a majority of the 14 paired fecal and urine specimens, supporting the similarity of urine and gut *E. coli* strains from the same subjects (Fig. 2e). In the 14 urine samples, we then evaluated for the presence of common and clinically significant classes of beta-lactamase genes: CTX-M, TEM, SHV, CMY, DHA, and OXA and correlated these results to antimicrobial susceptibility testing results from the uropathogen isolated in culture (Supplementary Fig. 5). We detected TEM genes, which confer ampicillin resistance, in 12 of the 13 urine samples that yielded ampicillin-resistant *E. coli* in culture. Furthermore, we detected CTX-M genes, which confer ampicillin and ceftriaxone resistance in all three urine specimens that yielded ampicillin- and ceftriaxone resistant *E. coli* in culture.

**Antibiotics and gut *Escherichia* and *Enterococcus* abundance.** We evaluated whether the most common antibiotics used in the cohort were risk factors for development of 1% relative gut abundance of *Escherichia* or 1% relative gut abundance of *Enterococcus* with the class of antibiotic administered as a time-dependent covariate in Cox regression analysis. The list of antibiotics given in the cohort is found in Supplementary Table 8. Beta-lactams had an increased hazard ratio for development of 1% relative *Enterococcus* gut abundance (HR: 4.6, $P < 0.001$) and cefazolin preoperative prophylaxis had a decreased hazard ratio for development of 1% relative *Escherichia* gut abundance (HR: 0.4, $P = 0.03$) (Supplementary Table 9).

## Discussion

The current study supports the notion that uropathogenic bacteria originate from the gut and further highlights a novel relationship between the gut microbiota and future development of bacteriuria and UTI in kidney transplant recipients. Prior studies have investigated the pathogenesis of UTI by assessing the mechanisms for survival and propagation of uropathogens in the urinary tract[9]. However, few studies have investigated the step preceding development of bacteriuria/UTI, the seeding of the urinary tract. In a pilot study, we evaluated the serial microbiome profiles in 26 kidney transplant recipients and reported that gut *Enterococcus* abundance was associated with *Enterococcus* UTIs[10], but we were not able to assess whether gut *Enterococcus* abundance was a risk factor for future development of *Enterococcus* bacteriuria or UTI.

In this study, we found that the gut abundances of *Escherichia* and *Enterococcus* are associated with future development of *Escherichia* and *Enterococcus* bacteriuria, respectively, independent of clinical factors like gender. We further found that the gut abundance of *Escherichia* was associated with symptomatic *Escherichia* UTI development. We did not find a similar relationship between the gut abundance of *Enterococcus* and *Enterococcus* UTI, but it is possible that we did not have sufficient numbers since only 14 of the 36 *Enterococcus* bacteriuria episodes were classified as symptomatic *Enterococcus* UTI.

To further support the gut microbiota-UTI link, we demonstrate that the strains of *E. coli* and *E. faecalis* in the urine specimens are most similar to the strains of *E. coli* and *E. faecalis* in fecal specimens from the same subjects, respectively. It is customarily thought that uropathogens present in the gut are the source of UTI via periurethral contamination[9]. However, prior evidence to support this hypothesis is limited. In fact, a recent study found that women with *E. coli* UTIs had different *E. coli* strains in their urine and in their stool using pulsed-field gel electrophoresis[23]. Literature investigating the role of uropathogenic genes like *FimH* in increasing the propensity for gut *E. coli* strains to cause UTI is also relatively limited[24]. Recent studies have found increased gut abundance of *Escherichia* or *Enterococcus* in patients with UTI compared to patients without UTIs[10,11], but they did not evaluate temporal dynamics of the gut microbiota. Our study expands upon this idea by demonstrating that increased fecal abundance of *Escherichia* and *Enterococcus* is associated with an increased risk of developing future bacteriuria with the respective organism. We also evaluated for the presence of uropathogenic genes like *FimH* and antibiotic resistance genes in paired *E. coli* urine and fecal specimens. We found the presence of these genes in both the urine and fecal specimens from the same subject in a majority of the samples, further supporting the strain alignment analysis that the *E. coli* in the gut is similar to the *E. coli* in the urine from the same subject. Interestingly, we did detect uropathogenic genes in asymptomatic *E. coli* bacteriuria cases. It is possible that the uropathogenic genes are present but not actively transcribed in these cases, that the presence of these gene products does not necessarily lead to symptomatic UTI, or that the cases were actually symptomatic but were not evaluated because the presence of urinary symptoms was not prospectively assessed.

Our study also investigated antibiotic usage associated with increased fecal *Escherichia* and *Enterococcus* abundances. We found that beta-lactams increased the risk for development of greater than 1% relative gut abundance of *Enterococcus*. *Enterococcus*, particularly *Enterococcus faecium*, is inherently resistant to many antibiotics in this class, allowing expansion of *Enterococcus* in the setting of the depletion of other gut bacteria by these antibiotics. Our current data support the notion that antibiotics can alter the gut microbiota and increase abundances of pathogenic bacteria in the gut. Recent studies have reported that these dysbiotic states with pathogenic bacteria can lead to infectious complications beyond *C. difficile* infection. Taur et al.

investigated the gut microbiome profiles of 94 bone marrow transplant recipients and reported that elevated gut *Enterococcus* abundances were associated with future development of *Enterococcus* bacteremia and that elevated gut Proteobacteria abundances were associated with future development of Proteobacteria bacteremia[25]. In a study of 28 non-Hodgkin lymphoma patients, Montassier et al. reported that gut microbial diversity was a predictor of future development of bloodstream infections[26].

The gut microbiota's relationship with bacteriuria development highlights a potential for novel gut microbiota-focused treatments. There are case reports that supports the notion that FMT could be a potential treatment for recurrent UTIs. In a case report of a heart and kidney transplant recipient who developed multiple events of *Enterococcus* bacteremia and UTIs as well as recurrent *C. difficile* infections, FMT decreased the fecal abundance of *Enterococcus* in the gut and led to resolution of *Enterococcus* infections in the subject for up to 14 months[27]. In a case series of FMT for recurrent *C. difficile* infections in a non-transplant population, the number of UTIs in the year after FMT was significantly lower than the number of UTIs in the year prior to FMT[28]. In this case series, however, the fecal microbiome was not profiled prior to and after FMT.

An important consideration in this study is the potential for cross-contamination of urine and fecal specimens. However, we took several steps to minimize its impact. With respect to urine contamination of fecal specimens, we collected fecal specimens using the Fisherbrand Commode Specimen Collection kit which was used in the Human Microbiome Project[29] and which, by design, minimizes cross-contamination of urine in fecal specimens[30]. We also highlight the significant difference in bacterial biomass between fecal and urine specimens by several log folds[31], making urine contamination of fecal specimens unlikely to affect our analyses. With respect to fecal contamination of urine specimens, subjects were asked to provide clean-catch midstream urine specimens as part of routine clinical care. We further highlight that our analyses also focused on evaluating the relationship between the gut microbiota and UTI. UTI is unlikely a contamination event because subjects experienced concomitant symptoms with the positive urine cultures and so this analysis is less likely to be affected by this type of contamination. Our study was also designed as a prospective collection of fecal specimens and fecal specimens were collected prior to the development of bacteriuria or UTI, making the collection processes independent of each other. Despite these steps taken to minimize cross-contamination, it is important to interpret the results of the study with due caution because cross-contamination of urine and fecal specimens is a potential confounder.

In addition, with respect to strain analysis, we did not have access to the urinary strains that were isolated for routine clinical care by the clinical microbiology laboratory. Unlike bloodstream isolates, urinary isolates are not saved for extended periods of time. However, we were able to utilize a computational algorithm (StrainPhlAn) to reconstruct consensus strains to evaluate the relatedness of the strains. While this approach provides an assessment of the genetic relatedness of the urine and fecal strains from the same subject, it is limited by the reconstruction of consensus strains from short base pair reads and may not reflect the full sequences of the strains that would be analyzable if the bacterial isolates were available. We also did not have urine from the donor to evaluate for the presence of donor-derived infections. However, of the 14 cases of *E. coli* bacteriuria for which strain analysis was performed, eight had a urine culture that was negative prior to the diagnosis of bacteriuria, suggesting the subjects developed the infection de novo. With respect to multivariable analysis, we included many variables that are associated with UTI in kidney transplant recipients, but we were not able to

include acute rejection or prolonged foley catheter utilization because of the low number of events as well as presence of a ureteral stent since all subjects had a ureteral stent placed. We performed a retrospective chart review of symptoms to distinguish symptomatic UTI from asymptomatic bacteriuria; since this was not a prospective collection of symptoms, it is possible that we did not capture all cases of symptomatic UTIs. Finally, we were not able to assess for changes in gut microbiota prior to transplantation and after transplantation since we only had five subjects who provided specimens prior to transplantation.

Despite these limitations, our data support uropathogenic gut abundance as a risk factor for development of bacteriuria and UTI. Our results further support future studies on modulating the gut microbiota as a potential novel strategy for preventing UTIs, especially in cases of recurrent UTIs.

## Methods

**Kidney transplant cohort.** Two-hundred eighty kidney transplant recipients who received kidney transplants during August 2015 to November 2016 were consented for serial collection of fecal and urine specimens, and 168 kidney transplant recipients provided at least 1 fecal specimen. The Weill Cornell Institutional Review Board approved this study and each subject gave written informed consent. No organs were procured from prisoners and all kidney transplant recipients were performed at NewYork-Presbyterian Hospital–Weill Cornell Medical Center. Urine specimens were analyzed at the clinical microbiology laboratory at NewYork-Presbyterian Hospital–Weill Cornell Medical Center. Each urine specimen was inoculated onto tryptic soy agar with sheep blood (Becton, Dickinson and Company [BD], Franklin Lakes, NJ) and MacConkey agar (BD) using a 1 microliter inoculation loop and incubated in ambient air at 35 °C. Bacteriuria was defined as having a positive urine culture (≥10,000 colony-forming units of a potential uropathogen per mL of urine) in the first 6 months after transplantation (up to 210 days post-transplantation). Clinical UTI was distinguished from bacteriuria by having bacteriuria and symptoms including: dysuria, frequency, urgency, or fevers[3] in the first 6 months after transplantation; the symptoms were determined by chart review by a single individual. Bacterial isolates recovered in culture were identified and their antimicrobial susceptibility profiles were determined using standard identification and broth microdilution antimicrobial susceptibility testing platforms. Antimicrobial susceptibility data were interpreted using the Clinical and Laboratory Standards Institute M100 document depending on the year of the culture: M100-S25[32] (2015) or M100-S26[33] (2016). An a priori sample size determination was not performed given the limited preliminary data examining the relationship between gut microbial abundance and bacteriuria.

**Fecal specimen collections.** Subjects were given a Fisherbrand toilet specimen collection kit (Fisher Scientific, New Hampton, NH, USA) and were asked to provide a fecal specimen. As an inpatient, stool specimens were stored at 4 °C and as an outpatient, subjects stored the specimen with ice packs. Fecal specimens were collected and aliquoted into approximately 200 mg aliquots and stored at −80 °C. Fecal specimens were collected at post-transplant week 1, week 2, week 4, and week 12 and during episodes of diarrhea as well as during episodes of UTIs. Analysis was restricted to the 510 fecal specimens within the first 3 months after transplantation (up to 120 days post-transplantation) except for two additional fecal specimens from kidney transplant recipients who had recurrent bacteriuria (These samples were used for strain analysis.) With respect to stool samples and routine urine samples being obtained at the same time, only 28 (5%) were collected at the same time during the clinical visit. In most cases, stool samples were brought from home ($n = 269$, 53%) or collected in the hospital ($n = 213$, 42%).

**DNA extraction, 16S rRNA gene amplification, and sequencing.** DNA extraction, 16S rRNA gene amplification, and deep sequencing were performed[34]. The team performing DNA extraction, 16S rRNA gene amplification, and deep sequencing was blinded to the Bacteriuria Group status of the specimens. In brief, DNA was extracted from a frozen stool aliquot (<500 mg) using a solution containing 500 μL of extraction buffer (200 mM TrisHCl [pH 8.0]/200 mM NaCl/20 mM EDTA), 500 μL of phenol:chloroform:isoamyl alcohol (25:24:1), 200 μL of 20% SDS, and 500 μL of 0.1-mm-diameter zirconia/silica beads (BioSpec Products, Bartlesville, OK, USA). Mechanical disruption of the stool specimen was performed using a bead beater for 2 min, followed by two more rounds of phenol:chloroform: isoamyl alcohol extraction. DNA was isolated with ethanol precipitation and was resuspended in 50 μL of 1X TE buffer with 100μg/mL RNase. The DNA was further purified using QIAamp mini spin columns (Qiagen, Hilden, Germany). For each sample, duplicate 50 μL PCR reactions were performed, each containing 50 ng of purified DNA, 0.2 mM dNTPs, 1.5 mM MgCl₂, 2.5 U Platinum Taq DNA polymerase, 2.5 μL of 10 × PCR buffer, and 0.5 μM of each primer designed to amplify the 16S rRNA gene V4–V5 region: 563 F (5′-nnnnnnnn-NNNNNNNNNNNNN-AYTGGGYDTAAAGNG-3′) and 926 R (5′-nnnnnnnn-NNNNNNNNNNNNN-CC

GTCAATTYHTTTRAGT-3′). A unique 12-base Golay barcode (Ns) precedes primers for identification of the sample[35], and one to eight additional nucleotides precedes the barcode to offset the sequencing of the primers. Cycling conditions were 94 °C for 3 min, followed by 27 cycles of 94 °C for 50 s, 51 °C for 30 s, and 72 °C for 1 min. The final elongation step was at 72 °C for 5 min. Replicate PCRs were pooled, and the PCR products were further purified using the Qiaquick PCR Purification Kit (Qiagen, Hilden, Germany). PCR amplicons were quantified and pooled at equimolar amounts and were ligated with Illumina barcodes and adaptors using the Illumina TruSeq Sample Preparation protocol (Illumina Inc, San Diego, CA, USA). The completed library was sequenced on an Illumina Miseq platform with a paired end 250 bp × 250 bp kit.

**16S rRNA sequencing bioinformatics**. The 16S rRNA gene paired-end reads were merged and demultiplexed and run with the UPARSE[36] pipeline to: (a) perform error filtering, using maximum expected error[37], (b) group sequences into operational taxonomic units (OTUs) (97% distance-based similarity), and (c) identify and remove potential chimeric sequences, using de novo and reference-based methods. Taxonomic assignment was performed by using a custom Python script incorporating nucleotide BLAST[38], with NCBI RefSeq[39] as a reference training set. We utilized NCBI RefSeq as the training set as we have previously done[34,40,41] because it can capture species-level resolution with curation and manual inspection of the results and submission sources. A minimum E-value threshold of $1 \times 10^{-10}$ for assignments was utilized.

**Statistical analyses**. The distribution of continuous variables was compared using the two-tailed Wilcoxon rank sum test; the distribution of categorical variables was compared using two-tailed Fisher's exact test. A Cox Regression Hazard Model was used to estimate whether a relative fecal abundance of a specific organism was associated with development of bacteriuria of the same organism with relative fecal abundance as a time-ever dependent covariate where it was assumed that the abundance value was not crossed until the first time the microbial abundance value crossed the threshold. A multivariable Cox regression hazard model was performed to evaluate clinical variables we have previously analyzed[7] as well as the cause of end stage renal disease and calculated panel reactive antibody status. A similar Cox regression hazard model was used to estimate whether antibiotic class was associated with development of a relative fecal abundance of *Enterococcus* or *Escherichia* as a time-dependent covariate. All analyses were performed in R 3.3.3 with packages yingtools2 0.0.0.89, phyloseq 1.19.1, dplyr 0.7.7, tibble 2.1.1, reshape2 1.4.3, data.table 1.10.4–3, scales 0.5.0, Hmisc 4.1–1, stringr 1.4.0, stringi 1.4.3, biomformat 1.2.0, deplyr 0.7.7, coxphf 1.12, and ggtree 1.11.6.

**Strain analysis for paired urine and fecal specimens**. Urine specimens were also obtained serially in the kidney transplant recipients as part of a biobank in our center. Seventeen urine specimens had urine cultures that were positive for *E. coli* (*n* = 14), *E. faecalis* (*n* = 2), and *E. faecium* (*n* = 2) (one sample was positive for both *E. coli* and *E. faecalis*). Fecal specimens closest to the urine specimen and also had a relative gut abundance of greater than 2% of either *Escherichia* or *Enterococcus* were chosen as the paired specimens. The 2% cutoff was chosen so that we would have at least an estimated 5X genome coverage of *E. coli, E. faecalis*, and *E. faecium* for strain analysis.

The urine specimens were centrifuged at 2000 × *g* for 30 min on the same day of collection and the urine supernatant was stored at −80 °C in 1 mL aliquots. Cell-free DNA was extracted from the urine using the Qiagen Circulating Nucleic Acid Kit (Qiagen). Extracted cell-free DNA amounts were quantified with a Qubit Fluorometer using the HS dsDNA kit (Qubit, London, United Kingdom). A single-stranded library preparation method was used[13]. In brief, cell-free DNA was denatured and was ligated to biotinylated oligonucleotides. Primer extension was performed on streptavidin functionalized magnetic beads. Another set of adapters was ligated and the product was amplified with PCR (4–12 cycles). Characterization of the library was performed with the AATI fragment analyzer and samples were pooled and sequenced on an Illumina Next Seq Instrument (paired-end, 75 bp × 75 bp)[13]. Given potential low biomass, we included a control with template DNA (IDT-DNA synthetic oligo mix lengths 25, 40, 55, 70 bp; 0.20 μM eluted in TE buffer) in each sample batch.

The 17 stool samples underwent DNA extraction as described above and were quantified using a Qubit Fluorometer (Qubit). Sequencing libraries were prepared using an Illumina Tru Seq Kit (Illumina Inc, San Diego, CA, USA). Samples were pooled and sequenced on an Illumina Hi Seq 4000 instrument (paired-end, 100 bp × 100 bp).

We utilized kneaddata v.0.5.2 using bowtie2 v0.1[21] to trim paired fastq files and remove human sequences. Microbial sequence coverage was calculated by the reads after trimming and after decontamination of human sequences, multiplied by the fraction of the organism estimated by MetaPhlAn2 2.6.0[15], multiplied by the number of base pairs per read, and divided by the genome length of the organism. We then utilized StrainPhlAn 2.6.0[14] to obtain consensus strains for *E. coli, E. faecalis*, and *E. faecium*. MetaPhlAn2 2.6.0[15] was utilized to analyze the concatenated fastq file after kneaddata; the sample2markers script was utilized to obtain marker information for each sample; and generation of trees was constructed from alignments with StrainPhlAn script. Marker information for *E.*

*coli, E. faecalis*, and *E. faecium* was extracted from the MetaPhlAn2 database. The default relaxed 2 StrainPhlAn parameters were utilized (marker_in_clade 0.2, sample_in_marker 0.2, N_in_marker 0.8, gap_in_sample 0.8). The tree was further analyzed using a modified version of the StrainPhlAn ggtree R script.

**Uropathogenic/antibiotic resistance gene classification**. Concatenated fastq files after kneaddata were analyzed using HUMAnN2[19]. All bacterial genes were normalized to relative abundance using total sum scaling. A heatmap was generated for the paired *E. coli* urine and stool samples.

Concatenated fastq files after kneaddata were aligned using Bowtie2[21] sensitive mode to a MEGARES antibiotic resistance gene database[22]. The antibiotic resistance genes were normalized to reads per kilobyte per million (RPKM) and were grouped into classes of antibiotics. A heatmap was generated for the paired *E. coli* urine and stool samples.

**Reporting summary**. Further information on research design is available in the Nature Research Reporting Summary linked to this article.

## Data availability

All sequencing data as well as the deidentified clinical data that support the findings of this study have been uploaded to the database of Genotypes and Phenotypes (dbGaP) with accession number phs001879.v1.p1. Local institutional review board approval will be needed to access the sequencing data as well as the deidentified clinical data. The source data underlying Figs. 1, 2, Tables 1, 1, Supplementary Figs. 1–5, and Supplementary Tables 1–9 are provided as a Source Data file.

## Code availability

Custom python script for taxonomic assignment and R script for graphing phylogenetic tree can be found in a zip file in Supplementary Information.

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

## Acknowledgements

This research work was supported, in part, by the National Institutes of Health grants: K23AI124464 (J.R.L.), R21AI133331 (J.R.L., I.D.V.), and R37AI051652 (M.S.); by a Chinese American Medical Society Summer Research Fellowship (L.Z.); and by a Summer Research Fellowship from the Weill Cornell Clinical and Translational Science Center (N.A.S.).

## Author contributions

M.M. was involved in the conception of the study, in the 16S rRNA deep sequencing of fecal samples, in data analysis, and in writing of the manuscript. N.A.S., C.G., L.Z., E.E., J. H., and S.A. were involved in data collection and analysis and in writing of the manuscript. M.J.S., L.F.W., C.C., D.M.D., M.L., Y.T., and E.L. were involved in data analysis and in writing of the manuscript. L.L. was involved in the 16S rRNA gene deep sequencing of fecal samples. P.B. and I.D.V. were involved in the cell-free DNA sequencing of urine samples, data analysis, and in writing of the manuscript. E.P. and M. S. were involved in the conception of the study, in data analysis, and in writing of the manuscript. J.R.L. was involved in the conception of the study, in data analysis and collection, and in writing of the manuscript.

## Competing interests

The authors of this manuscript have the following competing interests to disclose. M.J.S. receives research support from Allergan, Merck, Contrafect, and BioFire Diagnostics, and received consulting fees from Acahogen, Inc. and Shionogi; L.F.W. receives research support from Accelerate Diagnostics, Inc and BioFire Diagnostics, LLC; C.C. receives support from Redhill, MERCK, Rebiotix, Seres, and Finch Therapeutics.; E.P. has received speaker honoraria from Bristol Myers Squibb, Celgene, Seres Therapeutics, MedImmune, Novartis and Ferring Pharmaceuticals and is an inventor on patent application #WPO2015179437A1, entitled Methods and compositions for reducing *Clostridium difficile* infection and #WO2017091753A1, entitled Methods and compositions for reducing vancomycin-resistant enterococci infection or colonization and holds patents that receive royalties from Seres Therapeutics, Inc.; J.R.L. receives research support from BioFire Diagnostics, LLC. D.D., P.B., I.D.V., M.S., and J.R.L. are inventors on patent application #W02018187521A2 entitled Methods of detecting cell-free DNA in biological samples.

## Additional information

**Peer Review Information** *Nature Communications* thanks Tessa Andermann, Tara Sigdel, and other, anonymous, reviewer(s) for their contribution to the peer review of this work. Peer reviewer reports are available.

