## [Peer Review File · Nature Communications]

Reviewers' comments:

Reviewer #2 (Remarks to the Author):

The authors have tried to address the comments raised by the reviewers in a thoughtful manner. A major criticism still lies in the inherent nature of the study design- particularly the following:

1. The cross contamination of urine and fecal matter is highly likely, especially given that the detection of bacterial species is by sequencing whereby even small amount of contamination will be detected. In fact this is the commonest method of ascending infection for cystitis, UTI and pyelonephritis. Hence the independence of these events is a difficult matter to convince the reader.
2. The authors actually minimize the significance of their hypotheses as they now state that "there was no significant difference in the abundance of these genes between symptomatic E. coli UTI and asymptomatic E. coli bacteriuria (Supplemental Figure 3)."
3. further when they were asked to explain "What was the outcome of the kidney transplant patients without UTI but with the same fecal microbiota as the group with the UTI? Their Response was: The gut microbiome is highly individualized and it is difficult to group subjects with the same fecal microbiota so we were unable to perform this analysis." This is actually a critical question to answer and show that indeed the fecal microbiota does or does not independently drive outcomes

Reviewer #3 (Remarks to the Author):

The updated manuscript is much improved and for the most part the authors have clearly justified their assumptions. I still have some concerns about the strain similarity component.

Pg 9. "Five of the 6 E. coli urine strains were closely related to the E. coli strains from the fecal specimens from the same subject".

This section still lacks a definition for "closely related". Is this just based on finding the nearest neighbor to each urine strain? If so, is this meaningful if the distances are larger than seen between independent hosts in some cases? Please specify exactly the criteria being used to identify the 5/6 pairs - subjectively one could easily argue that only 3 of these pairs are 'close' (especially given the tree is based on a set of markers rather than a whole genome). Similarly, the E. faecalis stool-urine distances are greater than intra-patient distances between stool samples.

Pg 4. "...future association with bacteriuria and UTI" - perhaps better "...investigate the associations between the gut microbiota and the risk of developing bacteriuria or UTI

Fig 1 & p6: specify 'relative abundance' rather than 'abundance'

Fig 3 caption: log scale rather than "logged scaled"

Why Humann2 for uropathogenic gene detection (bowtie vs database is used for ARG detection)

The authors find that increased relative abundances of either Enterococcus or Escherichia in the gut is a risk factor for bacteriuria, but only E. coli is a risk factor for symptomatic UTI. Can the authors speculate about why this might be the case? Is it expected given prior knowledge?

StrainPhlan is still a fairly esoteric tool; would be helpful if the authors could include a very short summary of how it works i.e., how does it determine strains? What is "distance" in Figure 2?

Pg 10. "We evaluated the following classes of antibiotic resistant genes: beta-lactams, fosfomycin...."

Authors should revise to make their list about resistance genes (e.g., beta-lactamases, etc.) or to e.g., "We evaluated genes conferring resistance to the following classes of antibiotics: beta-lactams, fosfomycin...."

Pg 10. "trimethoprim resistant genes" -> "trimethoprim resistance genes"

Figure 2 - what does "distance" represent in this figure? Should add x-axis title to figure.

Pg 31. "Among 6 paired urine-fecal specimens, the E. coli strains in the 5 urine specimens are phylogenetically closer to the 5 E. coli strains in the fecal specimens from the same Subjects."

Incomplete sentence- phylogenetically closer than what?

Figure 2 - check coloring for Subject 106

Consider combining Figure 2 and 3 to more easily connect concordance with StrainPhlan calculated relationships and gene presence/absence predictions.

Reviewer #4 (Remarks to the Author):

The authors have very nicely addressed almost all of the concerns brought up in the first review of the manuscript. The remaining questions include the following:

1) In the response to major comment #3 in the first review, the authors state that they made clear the difference between bacteriuria and clinical UTI. In the methods, please state how the determination of symptoms indicative of UTI were made (e.g. based on a retrospective chart review of notes by a single person or X number of people vs. based on prospective collection of symptoms).

2) In the response to major comment #9, the authors explained to the reviewer the choice of reference databases, however readers familiar with 16S will also need to know why RefSeq was chosen, given that this is an unusual choice

3) In the response to major comment #14, the authors note that they have only included the phylogenetic tree and have taken out the 100bp representative SNV differences. I believe that StrainPhlan will also provide the number of SNVs per kilobase and these numbers, if available, should be included in a table comparing stool and urine from the same individual across the different individuals. Alternatively, other tools such as StrainSifter would output phylogenetic trees and SNVs per kilobase and may be an alternative option, and one that is not marker gene-based (Tamburini F, Andermann TM, et al, Nature Medicine, December 2018).

4) In the response to minor comment #4, the authors provided a very helpful table of organisms that were found in urine cultures. I think that the label "Bacteriuria" for each organism is not necessary and can be removed. Further, adding species would be helpful from a clinical perspective as Staph aureus and Staph saprophyticus have different implications on a urine culture.

5) In the response to minor comment #5, the authors cited their previous 2013 paper as an explanation for why variables were chosen for the Cox regression models. In the prior paper, prolonged foley use was cited as a significant variable that was not included in any of the models in this manuscript. The reasons for variable choices should be made explicit in the methods section either by reference or a brief explanation about how these variables were chosen. I am also wondering why the variables chosen were different between bacteriuria and UTI models? If there is a specific reason, this needs to be made clear; if there is not, I wonder why this is the case? Granted, there are small differences but nonetheless, I am curious about this.

6) In the response to minor comment #6, the authors state that scripts will be made available to the reviewer, but I am unable to find them or any reference to them.

Other comments and questions I have include the following:

1) On page 20, the authors state that fecal specimens with relative abundance >2% were chosen and this is not made clear why this would be the case. Was this for accuracy/sensitivity in SNV comparison? Or to make the phylogenetic tree more readable? Is this why some matched stool samples were not included in Figure 2? I believe this should be included in the manuscript and why this cutoff was chosen for inclusion.

2) On page 9, the authors discuss how many of the strains included in the phylogenetic trees were closely related or not based on Figure 2. However, subject 7 and 41 appear closer to one another than to their respective paired within-patient samples. This was not discussed but perhaps it should be for clarification. And was there enough E.coli in subject 48's stool for comparison but it just did not meet the criteria of 2%? Why it was excluded can be mentioned in the figure legend. If it was present but just below 2%, was it also similar?

Point by Point Response

Reviewer #2 (Remarks to the Author):

The authors have tried to address the comments raised by the reviewers in a thoughtful manner. A major criticism still lies in the inherent nature of the study design- particularly the following:

1. The cross contamination of urine and fecal matter is highly likely, especially given that the detection of bacterial species is by sequencing whereby even small amount of contamination will be detected. In fact this is the commonest method of ascending infection for cystitis, UTI and pyelonephritis. Hence the independence of these events is a difficult matter to convince the reader.

Response. The Reviewer raises a good point. There are two possible modes of contamination that are relevant for the present study. First, it is possible that the analysis of stool samples is affected by contamination of fecal matter by urine. To manage this mode of contamination, we collected fecal specimens using the Fisherbrand Commode Specimen Collection Kit. This kit was used in the Human Microbiome Project (Human Microbiome Project, *Nature* 486(7402):207, 2012) and fits securely under the toilet seat in the back of the seat and avoids urine contamination of fecal matter. In addition, we note the significance difference in bacterial biomass between fecal specimens and urine specimens. As described in Karstens et al. *Nat Rev Urol* 15(12):735, 2018, fecal specimens have a bacterial abundance of approximately 10^{11} per gram, whereas urine specimens have $<10^5$ colony forming units per milliliter. There are therefore several log fold differences in bacterial biomass, making it unlikely that urine contamination of fecal specimens has an appreciable effect on our analyses and conclusions. Second, the urine specimen could be contaminated by fecal matter during specimen collection. Urine specimens were collected as part of routine clinical care and subjects were instructed to provide clean-catch midstream specimens to minimize fecal contamination of urine. Importantly, our analyses also focused on UTI which is unlikely attributed to a contamination event. UTI was defined as a positive urine culture and the presence of symptoms associated with UTI and it is unlikely subjects had concomitant symptoms consistent with a UTI and contaminated their urine specimens. We also highlight that our study design was a prospective collection of fecal specimens. Fecal specimens were collected in many cases before the development of bacteriuria or UTI, making the collection processes independent from each other, which strengthens our analyses with regards to potential contamination compared to a cross-sectional study comparing the gut microbiota at the time of bacteriuria or UTI. In 13 of the 17 paired cases for strain analysis, the fecal specimen was not collected on the same day as the urine specimen, again making the collection processes independent from each other. We have updated the discussion section to reflect the nuances of the cross-contamination issue.

Discussion Section. An important consideration in the study is the possibility of cross-contamination of urine and fecal specimens. However, we took several steps to minimize its impact. With respect to urine contamination of fecal specimens, we collected fecal specimens using the Fisherbrand Commode Specimen Collection kit which was used in the Human Microbiome Project²⁹ and which, by design, minimizes cross contamination of urine in fecal specimens³⁰. We also highlight the significant difference in bacterial biomass between fecal and urine specimens by several log folds³¹, making urine contamination of fecal specimens unlikely to affect our analyses. With respect to fecal contamination of urine specimens, subjects were asked to provide clean-catch midstream urine specimens as part of routine clinical care. We further highlight that our analyses also focused on evaluating the relationship between the gut microbiota and UTI. UTI is unlikely a contamination event because subjects experienced concomitant symptoms with the positive urine cultures and so this analysis is less likely to be affected by this type of contamination. Our study was also designed as a prospective collection of fecal specimens and fecal specimens were collected prior to the development of bacteriuria or UTI, making the collection processes independent of each other.

2. The authors actually minimize the significance of their hypotheses as they now state that "there was no significant difference in the abundance of these genes between symptomatic *E. coli* UTI and asymptomatic *E. coli* bacteriuria (Supplemental Figure 3)."

Response. We believe that the main goal of our study was to examine the relationship between abundance of uropathogens in the gut and the development of bacteriuria and UTI. We believe that our data support the association between the gut relative abundance of *Escherichia* and development of *Escherichia* bacteriuria, the association between the gut relative abundance of *Escherichia* and development of *Escherichia* UTI, and the association between the gut relative abundance of *Enterococcus* and *Enterococcus* bacteriuria. With respect to the detection of uropathogenic genes during asymptomatic *E. coli* bacteriuria, it is possible that the uropathogenic genes may be present in asymptomatic cases but not be actively transcribed. In our study, we did find a trend towards significance that the abundances of 2 of the 5 uropathogenic genes studied, *CsgBAC* and *BarA*, were higher in the symptomatic *Escherichia* UTI than the asymptomatic *Escherichia* bacteriuria ($P=0.08$, $P=0.08$, respectively) (Supplemental Figure 3).

3. further when they were asked to explain "What was the outcome of the kidney transplant patients without UTI but with the same fecal microbiota as the group with the UTI? Their Response was: The gut microbiome is highly individualized and it is difficult to group subjects with the same fecal microbiota so we were unable to perform this analysis." This is actually a critical question to answer and show that indeed the fecal microbiota does or does not independently drive outcomes

Response. As mentioned in our prior response, we are unable to match subjects with the same fecal microbiota since each individual has distinct microbiota, but we can group subjects with a particular threshold of relative gut abundance, which is the approach we have taken. In our study, we grouped subjects with a greater or lower than 1% relative gut abundance of *Escherichia* or with a greater or lower than 1% relative gut abundance of *Enterococcus* and evaluated the outcomes, namely bacteriuria and UTI. Using Cox Regression analyses, we report that a 1% relative gut abundance of *Escherichia* is a risk factor for future development of *Escherichia* bacteriuria (HR: 2.8, $P=0.002$ by multivariable cox regression analysis) (Table 1) and that a 1% relative gut abundance of *Enterococcus* is a risk factor for future development of *Enterococcus* bacteriuria (HR: 2.3, $P=0.02$) (Table 2). We also report that 1% relative gut abundance of *Escherichia* is associated with future development of *Escherichia* UTI (HR: 2.8, $P=0.02$) (Supplemental Table 3). This is not to say that there is a one-to-one relationship between microbial composition and these outcomes, but our analysis does demonstrate a link between the gut microbiota and future development of bacteriuria and UTI. We believe that this is a novel and important finding that could inform the management of UTI in the future.

Reviewer #3 (Remarks to the Author):

The updated manuscript is much improved and for the most part the authors have clearly justified their assumptions. I still have some concerns about the strain similarity component.

Pg 9. "Five of the 6 *E. coli* urine strains were closely related to the *E. coli* strains from the fecal specimens from the same subject".

This section still lacks a definition for "closely related". Is this just based on finding the nearest neighbor to each urine strain? If so, is this meaningful if the distances are larger than seen between independent hosts in some cases? Please specify exactly the criteria being used to identify the 5/6 pairs - subjectively one could easily argue that only 3 of these pairs are 'close' (especially given the tree is based on a set of markers rather than a whole genome). Similarly, the *E. faecalis* stool-urine distances are greater than intra-patient distances between stool samples.

Response. We thank the Reviewer for the detailed review of our manuscripts and the comments to improve the manuscript. We have incorporated the suggestion to better quantify the phylogenetic distance between the strains. We have analyzed the SNVs per 100 base pairs using a distant matrix with a Kimura Two-Parameter distance among the consensus sequences (distmat from EMBOSS: The European Molecular Biology Open Software Suite) (Rice, P., Longden, I., Bleasby, A. Trends in Genetics 16 (6): 276, 2000). Supplemental Table 5 shows all of the SNVs per 100 base pairs among all of the 20 *E. coli* urine and stool strains; Supplemental Table 6 shows all of the SNVs per 100 base pairs among the 10 *E. faecalis* urine and stool strains; Supplemental Table 7 shows all of the SNVs per 100 base pairs among the 5 *E. faecium* urine and stool strains. We now report in a more quantitative way that 5 of the 6 *E. coli* urine strains were closest to the *E. coli* strains from the fecal specimens from the same subjects, that both of the *E. faecalis* urine strains were closest to the *E. faecalis* strains from the fecal specimens from the same subjects, and that the *E. faecium* urine strain was closest to the *E. faecium* strain from the fecal specimen from the same subject. We have updated the results section.

Results Section. We utilized StrainPhlAn¹⁴ to assess the relatedness of the strains in the urine and in the stool. In brief, StrainPhlAn maps sequences from a specimen against the MetaPhlAn2 marker database¹⁵ to produce a consensus-marker sequence for species in the sample. StrainPhlAn then utilizes MUSCLE¹⁶ to align consensus sequences from samples to produce a multiple sequence alignment and utilizes RAXML¹⁷ to build the phylogenetic tree. Importantly, post-processing operations are utilized at each step to ensure high quality consensus sequences. In order to calculate differences at the base pair level, we utilized distmat from the European Molecular Biology Open Software Suite¹⁸ which measures the number of single nucleotide variants (SNVs) per 100 base pairs.

Utilizing StrainPhlAn, we obtained 20 consensus strains of *E. coli* among all of the 34 urine and fecal specimens among which there were 6 pairs of *E. coli* urine and stool from the same subject. Fig. 2a shows the related phylogeny of the *E. coli* consensus strains and that 5 of the 6 *E. coli* urine strains were closely related to the *E. coli* strains from the fecal specimens from the same subject. Using the SNVs per 100 base pair measurement, the closest strains for Subject 54 Urine 2, Subject 36 Urine, Subject 41 Urine, Subject 106 Urine, and Subject 165 Urine were the strains from the subjects' own stool samples (Supplemental Table 5). The closest strain for Subject 7 Urine was the strain from Subject 41 Urine but very closely similar to the strain from Subject 7 Stool (Supplemental Table 5).

We obtained 10 consensus strains of *E. faecalis* among all of the 34 urine and fecal specimens among which there were 2 pairs of *E. faecalis* urine and stool from the same subject. Fig. 2b shows the related phylogeny of the *E. faecalis* consensus strains and that both *E. faecalis* urine strains were closely related to the *E. faecalis* strains from the fecal specimens from the same subject. Using the SNVs per 100 base pair measurement, the closest strains for Subject 4 Urine and Subject 54 Urine 2 were the strains from the subjects' own stool samples (Supplemental Table 6).

We obtained 5 consensus strains of *E. faecium* among all of the 34 urine and fecal specimens among which there was 1 pair of *E. faecium* urine and stool from the same subject. Fig. 2c shows the related phylogeny of the *E. faecium* consensus strains and that the *E. faecium* urine strain was closely related to the *E. faecium* strain from the fecal specimen from the same subject. Using the SNVs per 100 base pair measurement, the closest strains for Subject 54 Urine 1 was the strain from the subject's own stool sample (Supplemental Table 7).

Pg 4. "...future association with bacteriuria and UTI" - perhaps better "...investigate the associations between the gut microbiota and the risk of developing bacteriuria or UTI

Response. We have updated the manuscript with the improved wording suggested by the Reviewer.

Fig 1 & p6: specify 'relative abundance' rather than 'abundance'

Response. We have made the changes to relative abundance.

Fig 3 caption: log scale rather than "logged scaled"

Response. We have corrected the errors.

Why Humann2 for uropathogenic gene detection (bowtie vs database is used for ARG detection)

Response. We utilized HUMAnN2 for uropathogenic gene detection because HUMAnN2 is a comprehensive pipeline that also utilizes Bowtie2 as we have done for ARG detection but can also annotate gene family abundance, pathway abundance, and pathway coverage via a reference database like UniRef90. Since UniRef90 includes uropathogenic genes, we utilized Humann2 for identification. However, UniRef90 does not include a complete antibiotic resistance gene database so we did not utilize Humann2 for ARG detection. We utilized bowtie2 (same alignment program as HUMAnN2) for direct alignment against the MEGARES database for ARG detection.

The authors find that increased relative abundances of either *Enterococcus* or *Escherichia* in the gut is a risk factor for bacteriuria, but only *E. coli* is a risk factor for symptomatic UTI. Can the authors speculate about why this might be the case? Is it expected given prior knowledge?

Response. We think that this may be related to the number of cases of symptomatic *Enterococcus* UTI. While we had 36 cases of *Enterococcus* bacteriuria, we only had 14 cases of symptomatic *Enterococcus* UTI. We report that there is not a relationship between the relative gut abundance of *Enterococcus* and *Enterococcus* UTI but the Hazard Ratio is positive at 1.8 with a P value of 0.28.

StrainPhIn is still a fairly esoteric tool; would be helpful if the authors could include a very short summary of how it works i.e., how does it determine strains? What is "distance" in Figure 2?

Response. We thank the Reviewer for the comment. We have included a brief overview of what StrainPhIn does in the Results Section (See below). The distance in Figure 2 represents the proportion of sequences that are different between the consensus strains (i.e. 0.01 on the axis represents 1% of the sequences of the consensus strains are different). We have updated the Figure and Figure legend.

Results Section. We utilized StrainPhIn¹⁴ to assess the relatedness of the strains in the urine and in the stool. In brief, StrainPhIn maps sequences from a specimen against the MetaPhlAn2 marker database¹⁵ to produce a consensus-marker sequence for species in the sample. StrainPhIn then utilizes MUSCLE¹⁶ to align consensus sequences from samples to produce a multiple sequence alignment and utilizes RAXML¹⁷ to build the phylogenetic tree. Importantly, post-processing operations are utilized at each step to ensure high quality consensus sequences. In order to calculate differences at the base pair level, we utilized distmat from the European Molecular Biology Open Software Suite¹⁸ which measures the number of single nucleotide variants (SNVs) per 100 base pairs.

Pg 10. "We evaluated the following classes of antibiotic resistant genes: beta-lactams, fosfomycin...."
Authors should revise to make their list about resistance genes (e.g., beta-lactamases, etc.) or to e.g., "We evaluated genes conferring resistance to the following classes of antibiotics: beta-lactams, fosfomycin...."

Response. We have made the suggested change.

Pg 10. “trimethoprim resistant genes” -> “trimethoprim resistance genes”

Response. Thank you; we have made the correction.

Figure 2 - what does “distance” represent in this figure? Should add x-axis title to figure.

Response. We have made the suggested change to the x axis. The distance in Figure 2 represents the proportion of sequences that are different between the consensus strains (i.e. 0.01 on the axis represents 1% of the sequences of the consensus strains are different). We have updated the Figure and Figure legend.

Pg 31. “Among 6 paired urine-fecal specimens, the *E. coli* strains in the 5 urine specimens are phylogenetically closer to the 5 *E. coli* strains in the fecal specimens from the same Subjects.”

Incomplete sentence- phylogenetically closer than what?

Response. We have revised the results section as we now report the distance between strains using SNVs per 100 base pairs using distmat from the EMBOSS: The European Molecular Biology Open Software Suite (Rice, P., Longden, I., Bleasby, A. Trends in Genetics 16 (6): 276, 2000).

Results Section. Utilizing StrainPhlAn, we obtained 20 consensus strains of *E. coli* among all of the 34 urine and fecal specimens among which there were 6 pairs of *E. coli* urine and stool from the same subject. Fig. 2a shows the related phylogeny of the *E. coli* consensus strains; 5 of the 6 *E. coli* urine strains were closely related to the *E. coli* strains from the fecal specimens from the same subject. Using the SNVs per 100 base pair measurement, the closest strains for Subject 54 Urine 2, Subject 36 Urine, Subject 41 Urine, Subject 106 Urine, and Subject 165 Urine were the strains from the subjects’ own stool samples (Supplemental Table 5). The closest strain for Subject 7 Urine was the strain from Subject 41 Urine but very closely similar to the strain from Subject 7 Stool (Supplemental Table 5).

We obtained 10 consensus strains of *E. faecalis* among all of the 34 urine and fecal specimens among which there were 2 pairs of *E. faecalis* urine and stool from the same subject. Fig. 2b shows the related phylogeny of the *E. faecalis* consensus strains and that both *E. faecalis* urine strains were closely related to the *E. faecalis* strains from the fecal specimens from the same subject. Using the SNVs per 100 base pair measurement, the closest strains for Subject 4 Urine and Subject 54 Urine 2 were the strains from the subjects’ own stool samples (Supplemental Table 6).

We obtained 5 consensus strains of *E. faecium* among all of the 34 urine and fecal specimens among which there was 1 pair of *E. faecium* urine and stool from the same subject. Fig. 2c shows the related phylogeny of the *E. faecium* consensus strains and that the *E. faecium* urine strain was closely related to the *E. faecium* strain from the fecal specimen from the same subject. Using the SNVs per 100 base pair measurement, the closest strains for Subject 54 Urine 1 was the strain from the subject’s own stool sample (Supplemental Table 7).

Figure 2 - check coloring for Subject 106

Response. We have corrected this error.

Consider combining Figure 2 and 3 to more easily connect concordance with StrainPhlAn calculated relationships and gene presence/absence predictions.

Response. We have combined Figure 2 and 3 as suggested by the Reviewer into one figure.

Reviewer #4 (Remarks to the Author):

The authors have very nicely addressed almost all of the concerns brought up in the first review of the manuscript. The remaining questions include the following:

1) In the response to major comment #3 in the first review, the authors state that they made clear the difference between bacteriuria and clinical UTI. In the methods, please state how the determination of symptoms indicative of UTI were made (e.g. based on a retrospective chart review of notes by a single person or X number of people vs. based on prospective collection of symptoms).

Response. Thank you for the positive assessment of our revisions. In this revised version, we have updated the methods to reflect that the determination of dysuria, frequency, urgency, or fever was based on a retrospective chart review by a single person. This was not a prospective collection of symptoms and we have listed this as a (new) limitation in the Discussion section.

Methods Section. Clinical urinary tract infection was distinguished from bacteriuria by having bacteriuria and symptoms including: dysuria, frequency, urgency, or fevers³ in the first 6 months after transplantation; the symptoms were determined by chart review by a single individual.

Discussion Section. We performed a retrospective chart review of symptoms to distinguish symptomatic UTI from asymptomatic bacteriuria; since this was not a prospective collection of symptoms, it is possible that we did not capture all cases of symptomatic UTIs.

2) In the response to major comment #9, the authors explained to the reviewer the choice of reference databases, however readers familiar with 16S will also need to know why RefSeq was chosen, given that this is an unusual choice

Response. We have updated the methods section to include the rationale as to why we utilized NCBI Refseq as the training set.

Relevant Methods Section. We utilized NCBI RefSeq as the training set as we have previously done^{31,37,38} because it can capture species-level resolution with curation and manual inspection of the results and submission sources.

3) In the response to major comment #14, the authors note that they have only included the phylogenetic tree and have taken out the 100bp representative SNV differences. I believe that StrainPhlan will also provide the number of SNVs per kilobase and these numbers, if available, should be included in a table comparing stool and urine from the same individual across the different individuals. Alternatively, other tools such as StrainSifter would output phylogenetic trees and SNVs per kilobase and may be an alternative option, and one that is not marker gene-based (Tamburini F, Andermann TM, et al, Nature Medicine, December 2018).

Response. We thank the Reviewer for the suggestion and have analyzed the SNVs per 100 base pairs using a distant matrix with a Kimura Two-Parameter distance among the consensus sequences (distmat from EMBOSS: The European Molecular Biology Open Software Suite) (Rice, P., Longden, I., Bleasby, A. Trends in Genetics 16 (6): 276, 2000). Supplemental Table 5 shows all of the SNVs per 100 base pairs among all of the 20 *E. coli* urine and stool strains; Supplemental Table 6 shows all of the SNVs per 100 base pairs among the 10 *E. faecalis* urine and stool strains; Supplemental Table 7 shows all of the SNVs per 100 base pairs among the 5 *E. faecium* urine and stool strains. In the revised manuscript, we report in a more quantitative way that 5 of the 6 *E. coli* urine strains were closest to the *E. coli* strains from the fecal specimens from the same subjects, that both of the *E. faecalis* urine strains were closest to the *E. faecalis* strains from the fecal specimens from

the same subjects, and that the *E. faecium* urine strain was closest to the *E. faecium* strain from the fecal specimen from the same subject. We have updated the results section.

Results Section. We utilized StrainPhlAn¹⁴ to assess the relatedness of the strains in the urine and in the stool. In brief, StrainPhlAn maps sequences from a specimen against the MetaPhlAn2 marker database¹⁵ to produce a consensus-marker sequence for species in the sample. StrainPhlAn then utilizes MUSCLE¹⁶ to align consensus sequences from samples to produce a multiple sequence alignment and utilizes RAXMI¹⁷ to build the phylogenetic tree. Importantly, post-processing operations are utilized at each step to ensure high quality consensus sequences. In order to calculate differences at the base pair level, we utilized distmat from the European Molecular Biology Open Software Suite¹⁸ which measures the number of single nucleotide variants (SNVs) per 100 base pairs.

Utilizing StrainPhlAn, we obtained 20 consensus strains of *E. coli* among all of the 34 urine and fecal specimens among which there were 6 pairs of *E. coli* urine and stool from the same subject. Fig. 2a shows the related phylogeny of the *E. coli* consensus strains; 5 of the 6 *E. coli* urine strains were closely related to the *E. coli* strains from the fecal specimens from the same subject. Using the SNVs per 100 base pair measurement, the closest strains for Subject 54 Urine 2, Subject 36 Urine, Subject 41 Urine, Subject 106 Urine, and Subject 165 Urine were the strains from the subjects' own stool samples (Supplemental Table 5). The closest strain for Subject 7 Urine was the strain from Subject 41 Urine but very closely similar to the strain from Subject 7 Stool (Supplemental Table 5).

We obtained 10 consensus strains of *E. faecalis* among all of the 34 urine and fecal specimens among which there were 2 pairs of *E. faecalis* urine and stool from the same subject. Fig. 2b shows the related phylogeny of the *E. faecalis* consensus strains and that both *E. faecalis* urine strains were closely related to the *E. faecalis* strains from the fecal specimens from the same subject. Using the SNVs per 100 base pair measurement, the closest strains for Subject 4 Urine and Subject 54 Urine 2 were the strains from the subjects' own stool samples (Supplemental Table 6).

We obtained 5 consensus strains of *E. faecium* among all of the 34 urine and fecal specimens among which there was 1 pair of *E. faecium* urine and stool from the same subject. Fig. 2c shows the related phylogeny of the *E. faecium* consensus strains and that the *E. faecium* urine strain was closely related to the *E. faecium* strain from the fecal specimen from the same subject. Using the SNVs per 100 base pair measurement, the closest strains for Subject 54 Urine 1 was the strain from the subject's own stool sample (Supplemental Table 7).

4) In the response to minor comment #4, the authors provided a very helpful table of organisms that were found in urine cultures. I think that the label "Bacteriuria" for each organism is not necessary and can be removed. Further, adding species would be helpful from a clinical perspective as *Staph aureus* and *Staph saprophyticus* have different implications on a urine culture.

Response. We have removed Bacteriuria from each of the listed organisms. We have also added the resolution of the different species in Supplemental Table 8. We did find one subject did not develop *Corynebacterium spp* bacteriuria and have updated Supplemental Table 1 and Supplemental Figure 1 as well.

5) In the response to minor comment #5, the authors cited their previous 2013 paper as an explanation for why variables were chosen for the Cox regression models. In the prior paper, prolonged foley use was cited as a significant variable that was not included in any of the models in this manuscript. The reasons for variable choices should be made explicit in the methods section either by reference or a brief explanation about how these variables were chosen. I am also wondering why the variables chosen were different between bacteriuria and UTI models? If there is a specific reason, this needs to be made clear; if there is not, I wonder why this is the case? Granted, there are small differences but nonetheless, I am curious about this.

Response. We thank the Reviewer for pointing out the differences in the list of variables in this manuscript compared to our earlier publication. We have now revised the list of variables to include the variables included in our 2013 paper (Lee et al., Transplantation 96(2):131, 2013). We have also included variables that Reviewer 2 suggested to include such as cause of end stage renal disease and calculated panel reactive antibody status. In comparison to our 2013 paper, there were 4 variables that we did not include: Diabetes mellitus status, history of prior kidney transplantation, ureteral stent placement, and prolonged foley catheter. We have now included diabetes mellitus status and history of prior kidney transplantation. Ureteral stent placement was universally present in all subjects so it was excluded. Prolonged Foley catheter placement was also excluded because it was present in only 3 of the subjects. Given the number of events to the number of variables, we also revised the analyses and performed univariate Cox regression analysis on each of the characteristics and for factors with a P value < 0.10, we performed multivariate Cox Regression analysis as previously done in our 2013 paper. We also revised the analyses involving antibiotic group and *Escherichia* 1% relative abundance and *Enterococcus* 1% relative abundance (Supplemental Table 9).

Using the revised variables, we performed the Cox regression analyses for *Escherichia* and *Enterococcus* bacteriuria and UTI (Tables 1 and 2, Supplemental Tables 3 and 4). See updated results section below. In multivariable Cox regression analyses, we continue to report a significant association between *Escherichia* 1% gut relative abundance and *Escherichia* bacteriuria (HR: 2.8, P=0.002), between *Escherichia* 1% gut relative abundance and *Escherichia* UTI (HR: 2.8, P=0.02), and between *Enterococcus* 1% gut relative abundance and *Enterococcus* bacteriuria (HR: 2.3, P=0.02), significant differences that we previously found in the original model.

Methods Section. A multivariable Cox regression hazard model was performed to evaluate clinical variables that we have previously analyzed⁷ as well as the cause of end stage renal disease and calculated panel reactive antibody status.

Results Section. Using a univariate Cox Regression analysis with the relative gut abundance as a time-dependent covariate, we found that a 1% relative gut abundance of *Escherichia* was associated with future development of *Escherichia* bacteriuria (Hazard Ratio [HR]: 2.8, P = 0.002) (Table 1). In multivariate analysis that included the significantly associated univariate characteristics of female gender and cefazolin preoperative antibiotic prophylaxis, a 1% relative gut abundance of *Escherichia* continued to be associated with future development of *Escherichia* bacteriuria (HR: 2.8, P = 0.002) (Table 1). We also found that a 1% relative gut abundance of *Enterococcus* was associated with future development of *Enterococcus* bacteriuria (HR: 2.4, P = 0.01) (Table 2). In multivariate analysis that included the significantly associated univariate characteristics of deceased donor transplantation and delayed graft function, a 1% relative gut abundance of *Enterococcus* continued to be associated with future development of *Enterococcus* bacteriuria (HR: 2.3, P = 0.02) (Table 2).

We also analyzed the relationship between 1% relative gut abundance of *Escherichia* and future development of *Escherichia* UTI. In univariate analysis, a 1% relative gut abundance of *Escherichia* was associated with future development of *Escherichia* UTI (HR: 2.9, P=0.01) and in multivariable analysis controlling for gender, a 1% relative gut abundance of *Escherichia* continue to be associated with future development of *Escherichia* UTI (HR: 2.8, P=0.02) (Supplemental Table 3). In univariate analysis, we did not find a significant relationship between a 1% relative gut abundance of *Enterococcus* and future development of *Enterococcus* UTI (HR: 1.8, P=0.28) (Supplemental Table 4).

6) In the response to minor comment #6, the authors state that scripts will be made available to the reviewer, but I am unable to find them or any reference to them.

Response. We have updated the methods with a code availability statement and include the scripts in the Supplementary Methods (Please see zip file).

Other comments and questions I have include the following:

1) On page 20, the authors state that fecal specimens with relative abundance >2% were chosen and this is not made clear why this would be the case. Was this for accuracy/sensitivity in SNV comparison? Or to make the phylogenetic tree more readable? Is this why some matched stool samples were not included in Figure 2? I believe this should be included in the manuscript and why this cutoff was chosen for inclusion.

Response. We thank the Reviewer for the specific issues raised. We have now included the rationale for the inclusion criteria in the Results Section. Our goal was to compare the strains of *E. coli* in the fecal specimens to the strains of *E. coli* in the urine specimens and repeat this for *E. faecalis* and *E. faecium*. The 2% relative abundance was chosen based on the results from the 16S rRNA gene sequencing data so that we would have enough genome coverage of the organism for strain analysis. For example, if there was a paired sample and the *E. coli* fecal abundance was 0.001%, it would be difficult to generate enough sequences for strain analysis. With a criteria of a depth of 40 million raw sequences, 50% alignment to human sequences, 2% relative abundance of *E. coli*, and 100 base pair sequences per read, we estimated approximately 8X genome coverage for *E. coli* (5 million base pairs per *E. coli* genome) and 13X genome coverage for *E. faecalis* or *E. faecium* (3 million base pairs per *Enterococcus* genome) so that we would hopefully have at least 5X coverage. Of note, the 2% relative abundance criteria was not chosen for inclusion of the specimens for strain analysis, but selection for metagenomic sequencing. We have updated the Results section with the following:

Results Section. To assess the relatedness of strains in the gut and in the urine, we evaluated 17 fecal specimens that were paired to 17 urine specimens in which urine cultures were positive for *Escherichia coli* (n=14), *Enterococcus faecalis* (n=3), and *Enterococcus faecium* (n=1) (one urine culture was positive for both *E. coli* and *E. faecalis*) and in which relative gut abundance of *Escherichia* or *Enterococcus* in the gut was above two percent by 16S rRNA deep sequencing. The 2 percent cutoff was chosen so that we would have at least an estimated 5X genome coverage of *E. coli*, *E. faecalis*, and *E. faecium* for strain analysis based on an estimated 40 million reads per sample and 50% alignment to human sequences.

With respect to why some matched stool and urine samples were not included in the Figure, all matched urine samples and stool samples were included in the StrainPhlAn analysis. However, StrainPhlAn has post-processing analyses to ensure the obtainment of high quality consensus sequences. In brief, StrainPhlAn maps sequences from a specimen against the MetaPhlAn2 marker database to produce a consensus-marker sequence for species in the sample. StrainPhlAn then utilizes MUSCLE to align consensus sequences from samples to produce a multiple sequence alignment and utilizes RAXML to build the phylogenetic tree. Importantly, post-processing operations are utilized at each step to ensure high quality consensus sequences such as a percentage of ambiguous bases > 20% results in discarding the sequence. We have revised the results section to reflect that all samples were analyzed using StrainPhlAn.

Results Section. Utilizing StrainPhlAn, we obtained 20 consensus strains of *E. coli* among all of the 34 urine and fecal specimens among which there were 6 pairs of *E. coli* urine and stool from the same subject. Fig. 2a shows the related phylogeny of the *E. coli* consensus strains; 5 of the 6 *E. coli* urine strains were closely related to the *E. coli* strains from the fecal specimens from the same subject. Using the SNVs per 100 base pair measurement, the closest strains for Subject 54 Urine 2, Subject 36 Urine, Subject 41 Urine, Subject 106 Urine, and Subject 165 Urine were the strains from the subjects' own stool samples (Supplemental Table 5). The closest strain for Subject 7 Urine was the strain from Subject 41 Urine but very closely similar to the strain from Subject 7 Stool (Supplemental Table 5).

We obtained 10 consensus strains of *E. faecalis* among all of the 34 urine and fecal specimens among which there were 2 pairs of *E. faecalis* urine and stool from the same subject. Fig. 2b shows the related phylogeny of the *E. faecalis* consensus strains and that both *E. faecalis* urine strains were closely related to the *E. faecalis* strains from the fecal specimens from the same subject. Using the SNVs per 100 base pair measurement, the

closest strains for Subject 4 Urine and Subject 54 Urine 2 were the strains from the subjects' own stool samples (Supplemental Table 6).

We obtained 5 consensus strains of *E. faecium* among all of the 34 urine and fecal specimens among which there was 1 pair of *E. faecium* urine and stool from the same subject. Fig. 2c shows the related phylogeny of the *E. faecium* consensus strains and that the *E. faecium* urine strain was closely related to the *E. faecium* strain from the fecal specimen from the same subject. Using the SNVs per 100 base pair measurement, the closest strains for Subject 54 Urine 1 was the strain from the subject's own stool sample (Supplemental Table 7).

2) On page 9, the authors discuss how many of the strains included in the phylogenetic trees were closely related or not based on Figure 2. However, subject 7 and 41 appear closer to one another than to their respective paired within-patient samples. This was not discussed but perhaps it should be for clarification. And was there enough *E. coli* in subject 48's stool for comparison but it just did not meet the criteria of 2%? Why it was excluded can be mentioned in the figure legend. If it was present but just below 2%, was it also similar?

Response. We thank the reviewer for the initial suggestion to calculate the SNV/100 base pair distance matrix between all of the samples. We have performed this analysis and better quantify the distance between the strains (Supplemental Tables 5, 6, and 7). We now report that the Subject 7 Urine Strain was closest by SNV/100 base pair to Subject 41 Urine Strain but its second closest was to Subject 7 Stool Strain (See below Results). We apologize for the confusion about the 2% criteria. As described in the last response, the 2% criteria was chosen based on the 16S rRNA sequencing data to have sufficient coverage of the genome for strain analysis and we included all samples into StrainPhlAn analysis. We also note that Subject 48's sample in the Figure was stool. We apologize for the error and we have corrected this in the new Figure. We did not obtain a consensus sequence for Subject 48 urine after StrainPhlAn analysis. We have updated the Figure Legend to reflect that not all consensus sequences were obtained after StrainPhlAn analysis.

Results Section. We utilized StrainPhlAn¹⁴ to assess the relatedness of the strains in the urine and in the stool. In brief, StrainPhlAn maps sequences from a specimen against the MetaPhlAn2 marker database¹⁵ to produce a consensus-marker sequence for species in the sample. StrainPhlAn then utilizes MUSCLE¹⁶ to align consensus sequences from samples to produce a multiple sequence alignment and utilizes RAXML¹⁷ to build the phylogenetic tree. Importantly, post-processing operations are utilized at each step to ensure high quality consensus sequences. In order to calculate differences at the base pair level, we utilized distmat from the European Molecular Biology Open Software Suite¹⁸ which measures the number of single nucleotide variants (SNVs) per 100 base pairs.

Utilizing StrainPhlAn, we obtained 20 consensus strains of *E. coli* among all of the 34 urine and fecal specimens among which there were 6 pairs of *E. coli* urine and stool from the same subject. Fig. 2a shows the related phylogeny of the *E. coli* consensus strains; 5 of the 6 *E. coli* urine strains were closely related to the *E. coli* strains from the fecal specimens from the same subject. Using the SNVs per 100 base pair measurement, the closest strains for Subject 54 Urine 2, Subject 36 Urine, Subject 41 Urine, Subject 106 Urine, and Subject 165 Urine were the strains from the subjects' own stool samples (Supplemental Table 5). The closest strain for Subject 7 Urine was the strain from Subject 41 Urine but very closely similar to the strain from Subject 7 Stool (Supplemental Table 5).

We obtained 10 consensus strains of *E. faecalis* among all of the 34 urine and fecal specimens among which there were 2 pairs of *E. faecalis* urine and stool from the same subject. Fig. 2b shows the related phylogeny of the *E. faecalis* consensus strains and that both *E. faecalis* urine strains were closely related to the *E. faecalis* strains from the fecal specimens from the same subject. Using the SNVs per 100 base pair measurement, the closest strains for Subject 4 Urine and Subject 54 Urine 2 were the strains from the subjects' own stool samples (Supplemental Table 6).

We obtained 5 consensus strains of *E. faecium* among all of the 34 urine and fecal specimens among which there was 1 pair of *E. faecium* urine and stool from the same subject. Fig. 2c shows the related phylogeny of the *E. faecium* consensus strains and that the *E. faecium* urine strain was closely related to the *E. faecium* strain from the fecal specimen from the same subject. Using the SNVs per 100 base pair measurement, the closest strains for Subject 54 Urine 1 was the strain from the subject's own stool sample (Supplemental Table 7).

We have updated the competing interests section which is also listed below.

The authors of this manuscript have the following competing interests to disclose. M.J.S. receives research support from Allergan, Merck, Contrafect, and BioFire Diagnostics, and received consulting fees from Acahogen, Inc. and Shionogi; L.F.W. receives research support from Accelerate Diagnostics, Inc and BioFire Diagnostics, LLC; C.C. receives support from Redhill, MERCK, Rebiotix, Seres, and Finch Therapeutics.; E.P. has received speaker honoraria from Bristol Myers Squibb, Celgene, Seres Therapeutics, MedImmune, Novartis and Ferring Pharmaceuticals and is an inventor on patent application # WPO2015179437A1, entitled "Methods and compositions for reducing *Clostridium difficile* infection" and #WO2017091753A1, entitled "Methods and compositions for reducing vancomycin-resistant enterococci infection or colonization" and holds patents that receive royalties from Seres Therapeutics, Inc.; J.R.L. receives research support from BioFire Diagnostics, LLC. D.D., P.B., I.D.V., M.S., and J.R.L. are inventors on patent application #W02018187521A2 entitled "Methods of detecting cell-free dna in biological samples."

We believe that we have addressed the Reviewers' and Editor's thoughtful comments and that these comments have made this into a much improved manuscript. We thank you in advance for your further consideration of this manuscript for publication.

Sincerely,

John Lee, M.D., M.S.

Reviewers' comments:

Reviewer #3 (Remarks to the Author):

The authors have addressed my comments.

Reviewer #4 (Remarks to the Author):

The authors have very carefully and thoughtfully addressed all of the prior concerns. Their efforts to include all the pertinent information and make the changes as requested are very much appreciated.

I would recommend the inclusion of a supplementary figure demonstrating the timing of stool collections prior to the identification of at least the 17 urine specimens in which urine cultures were positive for E.coli or Enterococcus faecalis or faecium. I am curious as to which of these were UTIs and which were bacteriuria? You may want to label this, or include this in the text or figure legend.

Minor points:

1. Figure 2 labels are too small to read and will need to be enlarged
2. Figure 2d should include the function or putative functions of each of these genes
3. The antibiotic resistance noted in Figure 2e should be labeled with the gene that is responsible for this resistance either in the figure or in the legend
4. Line 315 should read “demonstrating that increased fecal abundance of E.coli and Enterococcus is associated with an increased risk of developing future...” as causality has not yet been established.

Reviewer #5 (Remarks to the Author):

On the first concern about the cross contamination of the urine and the fecal matter, the authors have tried to justify their observation being not impacted by such cross contamination. The statement added on the discussion in the revised manuscript highlights some possibility of such contamination which is better for the readers to know. I recommend that the authors list this as a potential confounder in the study despite their trying their best to avoid it.

On the second concern, the authors have reiterated that the main goal of the study was to examine the relationship between abundance of uropathogens in the gut and the development of bacteriuria and UTI. It would be better to delineate difference in between uropathogens and uropathogenic genes where relevant. The authors' response "With respect to the detection of uropathogenic genes during asymptomatic E. coli bacteriuria, it is possible that the uropathogenic genes may be present in asymptomatic cases but not be actively transcribed" needs to be discussed in the discussion section.

On the third concern, I think the authors rebuttal is acceptable in the context of the manuscript's title. The question the reviewer brought up is quite relevant but it may span outside the scope of the presented work/data by the manuscript. The rebuttal is therefore acceptable.

Point by Point Response

Reviewers' comments:

Reviewer #3 (Remarks to the Author):

The authors have addressed my comments.

Response. We thank the Reviewer for the previous comments that have enhanced the manuscript, especially with respect to strain analysis. Based on the Editor's comment, we have updated the discussion to reflect on the potential limitations of the StrainPhlAn tool especially since this tool creates consensus strains based on short base pair reads.

Discussion. In addition, with respect to strain analysis, we did not have access to the urinary strains that were isolated for routine clinical care by the clinical microbiology laboratory. Unlike bloodstream isolates, urinary isolates are not saved for extended periods of time. However, we were able to utilize a computational algorithm (StrainPhlAn) to reconstruct consensus strains to evaluate the relatedness of the strains. While this approach provides an assessment of the genetic relatedness of the urine and fecal strains from the same subject, it is limited by the reconstruction of consensus strains from short base pair reads and may not reflect the full sequences of the strains that would be analyzable if the bacterial isolates were available.

Reviewer #4 (Remarks to the Author):

The authors have very carefully and thoughtfully addressed all of the prior concerns. Their efforts to include all the pertinent information and make the changes as requested are very much appreciated.

Response. We thank the Reviewer for the very careful review of our manuscript and also for all of the previous comments that have improved the manuscript, especially with respect to strain analysis.

I would recommend the inclusion of a supplementary figure demonstrating the timing of stool collections prior to the identification of at least the 17 urine specimens in which urine cultures were positive for *E.coli* or *Enterococcus faecalis* or *faecium*. I am curious as to which of these were UTIs and which were bacteriuria? You may want to label this, or include this in the text or figure legend.

Response. We have included the supplemental figure (Supplemental Figure 3) and updated the results section as suggested by the Reviewer. In this figure, the timing of the collection of the stool specimens and urine specimens are presented along with the urine culture results and whether the urine cultures were associated with bacteriuria or UTI.

Supplemental Figure 3

Supplemental Figure 3. Timing of Collection of Fecal Specimens and Urine Specimens in the 17 Paired Cases for Strain Analysis. On the x axis is the day in relationship to when the clinical urine culture was obtained and on the y axis is the 17 cases for strain analysis. All urine specimens that underwent metagenomic sequencing were obtained on the same day as the urine culture, represented by an X shape. Fecal specimens that underwent metagenomic sequencing were obtained on the day represented by a triangle shape. The text inside the box on the left represents the clinical urine culture results as well as whether the urine culture was associated with bacteriuria or UTI. Source data are provided as a source data file.

Based on our review of the urine cultures, we revised the numbers of the urine cultures associated with *E. coli*, *E. faecalis*, and *E. faecium* and updated the text in the Results and Supplementary Note 1 section. In order to better visualize the bacteriuria/UTI status on the strain analysis, we added details of the urine status (bacteriuria or UTI status) in the strain analysis in Figure 2 (See Figure 2 on next page).

Results. To assess the relatedness of strains in the gut and in the urine, we evaluated 17 fecal specimens that were paired to 17 urine specimens in which urine cultures were positive for *Escherichia coli* (n=14), *Enterococcus faecalis* (n=2), and *Enterococcus faecium* (n=2) (one urine culture was positive for both *E. coli* and *E. faecalis*) and in which relative gut abundance of *Escherichia* or *Enterococcus* was above two percent by 16S rRNA gene sequencing. The timing of collection of stool specimens in relationship to the collection of urine specimens is shown in Supplemental Fig. 3.

Minor points:

1. Figure 2 labels are too small to read and will need to be enlarged

Response. We have updated Figure 2 with larger and clearer labels. We have also added which urine strains were associated with bacteriuria or UTI. A smaller version of the figure is presented.

Figure 2

2. Figure 2d should include the function or putative functions of each of these genes

Response. We have included the functions of each of the genes in the Results section.

Results. We evaluated the following uropathogenic genes associated with UTI²⁰: *FimH*, *PapG*, *CsgBAC*, *BarA*, and *UvrY*. *FimH* is a gene that is part of the *fim* operon that encodes type 1 pili, allowing for attachment to the bladder epithelium; *PapG* is a gene that is part of the *pap* operon that encodes the P-type pilus, allowing for attachment to the bladder epithelium; *CsgBAC* is part of the operons that coordinates curli formation, which can facilitate biofilm formation; *BarA* and *UvrY* are part of the two component signaling systems, which regulate metabolic pathways and allow for adaption in the urinary tract²⁰.

3. The antibiotic resistance noted in Figure 2e should be labeled with the gene that is responsible for this resistance either in the figure or in the legend

Response. We have included the details of the genes for each class in the Results section as well as a supplemental data table. Each class encompassed a different number of genes/operons from the MEGARES antibiotic resistant genes database (Lakin et al., *Nucleic Acids Res* 45:D574-D580, 2017).

Results. We evaluated genes conferring resistance to the following classes of antibiotics: beta-lactams, fosfomycin, glycopeptides, sulfonamides, and trimethoprim. Evaluation using the MEGARES antibiotic resistant genes database²² included 115 gene/operons in the beta-lactams class; 7 gene/operons in the fosfomycin class; 72 gene/operons in the glycopeptide class; 4 gene/operons in the sulfonamides class; and 11 gene/operons in the trimethoprim class (The full list of genes is provided in Supplemental Data 1).

4. Line 315 should read “demonstrating that increased fecal abundance of *E.coli* and *Enterococcus* is associated with an increased risk of developing future...” as causality has not yet been established.

Response. We thank the Reviewer for pointing out this error and we have corrected it in the text.

Reviewer #5 (Remarks to the Author):

On the first concern about the cross contamination of the urine and the fecal matter, the authors have tried to justify their observation being not impacted by such cross contamination. The statement added on the discussion in the revised manuscript highlights some possibility of such contamination which is better for the readers to know. I recommend that the authors list this as a potential confounder in the study despite their trying their best to avoid it.

Response. We thank the Reviewer for the thoughtful review of our manuscript. We agree with the reviewer and have listed the cross contamination of urine and fecal matter as a potential confounder.

Discussion. An important consideration in this study is the potential for cross-contamination of urine and fecal specimens. However, we took several steps to minimize its impact. With respect to urine contamination of fecal specimens, we collected fecal specimens using the Fisherbrand Commode Specimen Collection kit which was used in the Human Microbiome Project²⁹ and which, by design, minimizes cross contamination of urine in fecal specimens³⁰. We also highlight the significant difference in bacterial biomass between fecal and urine specimens by several log folds³¹, making urine contamination of fecal specimens unlikely to affect our analyses. With respect to fecal contamination of urine specimens, subjects were asked to provide clean-catch midstream urine specimens as part of routine clinical care. We further highlight that our analyses also focused on evaluating the relationship between the gut microbiota and UTI. UTI is unlikely a contamination event because subjects experienced concomitant symptoms with the positive urine cultures and so this analysis is less likely to be affected by this type of contamination. Our study was also designed as a prospective collection of fecal specimens and fecal specimens were collected prior to the development of bacteriuria or UTI, making the collection processes independent of each other. Despite these steps taken to minimize cross contamination, it is important to interpret the results of the study with due caution because cross-contamination of urine and fecal is a potential confounder.

On the second concern, the authors have reiterated that the main goal of the study was to examine the relationship between abundance of uropathogens in the gut and the development of bacteriuria and UTI. It would be better to delineate difference in between uropathogens and uropathogenic genes where relevant. The authors' response "With respect to the detection of uropathogenic genes during asymptomatic *E. coli* bacteriuria, it is possible that the uropathogenic genes may be present in asymptomatic cases but not be actively transcribed" needs to be discussed in the discussion section.

Response. We thank the Reviewer for the suggestion. We have added discussion on the presence of uropathogenic genes in asymptomatic bacteriuria cases.

Discussion. We also evaluated for the presence of uropathogenic genes like *FimH* and antibiotic resistance genes in paired *E. coli* urine and fecal specimens. We found the presence of these genes in both the urine and fecal specimens from the same subject in a majority of the samples, furthering supporting the strain alignment analysis that the *E. coli* in the gut is similar to the *E. coli* in the urine from the same subject. Interestingly, we did detect uropathogenic genes in asymptomatic *E. coli* bacteriuria cases. It is possible that the uropathogenic genes are present but not actively transcribed in these cases, that the presence of these gene products does not necessarily lead to symptomatic UTI, or that the cases were actually symptomatic but were not evaluated because the presence of urinary symptoms was not prospectively assessed.

On the third concern, I think the authors rebuttal is acceptable in the context of the manuscript's title. The question the reviewer brought up is quite relevant but it may span outside the scope of the presented work/data by the manuscript. The rebuttal is therefore acceptable.

Response. We thank again the Reviewer for the review and the helpful suggestions to improve the manuscript.

Sincerely,

John Lee, M.D., M.S.

REVIEWERS' COMMENTS:

Reviewer #3 (Remarks to the Author):

I have no further comments.

Reviewer #4 (Remarks to the Author):

The authors have very thoughtfully and thoroughly addressed all of my concerns and questions in their revised manuscript.

Reviewer #5 (Remarks to the Author):

The authors have addressed the concerns raised. Therefore, I recommend the manuscript for publication.

Tara Sigdel

Point by Point Response

Reviewers' Comments:

Reviewer #3 (Remarks to the Author):

I have no further comments.

Reviewer #4 (Remarks to the Author):

The authors have very thoughtfully and thoroughly addressed all of my concerns and questions in their revised manuscript.

Reviewer #5 (Remarks to the Author):

The authors have addressed the concerns raised. Therefore, I recommend the manuscript for publication.

Tara Sigdel

Response. We thank the Reviewers for the thorough review of our manuscript and for the many suggestions which have enhanced this manuscript.